

# Supraglacial lake drainage through gullies and fractures

Angelika Humbert[1,2], Veit Helm[1], Ole Zeising[1], Niklas Neckel[1], Matthias H. Braun[3], Shfaqat Abbas Khan[4], Martin Rückamp[5], Holger Steeb[6,7], Julia Sohn[1,8], Matthias Bohnen[9], and Ralf Müller[9]

[1]Alfred-Wegener-Institut Helmholtz-Zentrum für Polar- und Meeresforschung, Bremerhaven, Germany
[2]Faculty of Geosciences, University of Bremen, Bremen, Germany
[3]Friedrich-Alexander-Universität Erlangen-Nürnberg, Institut für Geographie, Erlangen, Germany
[4]DTU Space, National Space Institute, Technical University of Denmark, Department of Geodesy and Earth Observations, Copenhagen, Denmark
[5]Bavarian Academy of Sciences and Humanities, Geodesy and Glaciology, Munich, Germany
[6]Institute of Applied Mechanics (CE), University of Stuttgart, Germany
[7]SC SimTech, University of Stuttgart, Germany
[8]IU International University of Applied Sciences, Erfurt, Germany
[9]Division of Continuum Mechanics, Institute for Mechanics, Technical University of Darmstadt, Darmstadt, Germany

**Correspondence:** Angelika Humbert (angelika.humbert@awi.de)

**Abstract.** Supraglacial lake drainage through fractures delivers vast amounts of water to the ice sheet base on timescales of hours. This study is concerned with the mechanisms of supraglacial lake drainage and how a particular area of Nioghalvfjerds­bræ with a lake of a volume up to $1.23 \cdot 10^8 \, \mathrm{m}^3$. We found extensive fracture fields being formed and vertical displacement across the fracture faces in some instances. The fractures are accommodated with triangular gullies, in the order of 10's m's, into which water is flowing still weeks after the main lake drainage, but also instances in which the water level rises over the surface end of summer. These gullies are sometimes reactivated in subsequent years and their size at the surface remains unchanged over some years, which is in agreement with viscoelastic modelling. Using ice-penetrating radar, we find englacial, three-dimensional features originating from the drainage, changing over years but remaining detectable even years after their formation. The drained water forms a blister underneath the lake, which is released over several weeks. In this area, no lakes existed before an increase in atmospheric temperatures in the mid-1990s as we demonstrate using reanalysis data. It is trans­formed from lake-free to frequent, abrupt drainage delivering massive amounts of lubricant and freshwater at the seaward margin.

## 1 Introduction

Since the mid-1990s the Greenland Ice Sheet (GrIS) has been losing mass, with an acceleration of its contribution to sea level rise since the early 2000s (Shepherd et al., 2020). While the mass loss was most prominent in the southern and north-western parts in the first decade, it has now reached the Northeast of Greenland. Most prominent and latest is the disintegration of the floating tongue of Zacharias Isbræ (ZI) (Khan et al., 2014; Mouginot et al., 2015) in 2013. Since then, only three floating tongues are remaining, Petermann Glacier, Ryder Glacier and Nioghalvfjerdsbræ (79°N Glacier, 79NG), with the latter showing first signs of destabilisation (Humbert et al., 2023b). For floating tongue glaciers, oceanic heat is supposed to be a main driver



for thinning and stability (Straneo and Heimbach, 2013; Schaffer et al., 2020), but subglacial freshwater discharge is also playing a role in this system (Reinert et al., 2023; Zeising et al., 2024). The mass loss of GrIS is attributed to two contributions: (i) glacier acceleration and (ii) changes in surface mass balance, which primarily means an increase in surface melting (van den Broeke et al., 2016; Mouginot et al., 2019). Changes in surface mass balance are a consequence of changes in air temperatures. In the mid-1990's the weather station Danmarkshavn (76.8°N,18.7°W) recorded an increase in 2 m temperature of more than

1 °C (Zhang et al., 2022). Although the weather station is located roughly 300 km south of 79NG and ZI, the observations of changes in these glaciers correspond to this change in atmospheric forcing (AF), as discussed in Khan et al. (2014), Humbert et al. (2023b) and Zeising et al. (2024). In this study, we will address how the formation of a massive meltwater lake is linked to changes in surface temperatures.

The dynamics of ice sheets and outlet glaciers is gravity-driven lubricated flow. The surface runoff in the ablation zone partially

reaches the ice base to act as an additional lubricant. Meltwater that is locally formed in crevassed areas is transported through crevasses to the glacier base and can lead to a seasonal acceleration in ice flow (e.g. Rosenau et al., 2015; Vijay et al., 2019; Neckel et al., 2020). Further upstream and at higher elevation locations, the surface meltwater is taking two routes: percolation into the porous firn matrix or surface runoff. With a fast melt onset, the amount of meltwater cannot percolate entirely into the firn matrix. The large amounts of water that remain on the surface form runoff. The runoff becomes an organised water flow

system containing rivers and streams that lead water to topographic depressions in which supraglacial lakes are forming. These depressions correspond to basal topographic lows. The size of a supraglacial lake can range from a few hundred square metres to tens of square kilometres. We focus in this study on a single lake with a size of about 21 km$^2$, which we use as a case study. Supraglacial lakes may either drain straight into moulins, or partially by overflow, or by feeding lower elevation lakes via rivers and streams. Another type of drainage is through cracks and englacial channels (e.g. Das et al., 2008) on which this

study focuses. The mechanism of supraglacial lake drainage by fractures is not yet well understood. In some locations, the ice surface is already highly crevassed when moving into a depression. In this case hydrofracturing is the likely drainage mechanism. One example of this type is presented in Neckel et al. (2020). Detailed geophysical studies have revealed that drainage events are rapid and that fractures open to form a water passage and close after the main drainage took place (Das et al., 2008; Doyle et al., 2013; Stevens et al., 2015; Chudley et al., 2019).

In Greenland areas are likely to exist in which an englacial system forms during lake drainage. In contrast, in mountain glaciers, englacial channels form from tensile cracks at the surface or from shear cracks that arise at the lateral margins. In summer, the surface cracks were filled with meltwater. Due to creep closure, the tip of the water-filled crack becomes shut off. This mechanism forms englacial voids with mainly horizontal geometry and diameters of a few centimetres (Fountain and Walder, 1998). Fountain et al. (2005) observed a fracture network in Storglaciären covering up to 96% of the ice thickness with fractures

openings of 0.3–20 cm. In some of the fractures, water was moving with typical speeds of 1–2 cm s$^{-1}$. Fountain and coworkers used radio echo sounding to observe internal reflections arising from englacial channels, a technique we also apply in our study. They found steeply dipping features in radargrams, which were confirmed by hot-water drilling to be fractures. The study on Storglaciären suggests that the englacial hydraulic system is not dominated by tubular conduits but by interconnected fractures. A different study found vertical boreholes to intersect triangular-shaped conduits of 0.1 m diameter (Fountain and



Walder, 1998). All those findings point to englacial features arising from fractures rather than melting processes. In contrast to mountain glaciers, geometry and water transport through englacial channels in GrIS are only poorly understood. A substantial difference between the englacial networks in mountain glaciers and outlet glaciers in Greenland is potentially the spatial scale, as we will discuss in this work.

As fractures are suggested as the origin of englacial pathways, it is important to consider the material behaviour of ice. While
ice in glaciers was mainly treated as a viscous fluid obeying non-Newtonian rheology, in recent decades the solid nature of ice, hence the elastic contribution to deformation, has been explored in more detail (Reeh et al., 2000; Gudmundsson, 2011; Humbert et al., 2015; Christmann et al., 2019; Rosier and Gudmundsson, 2020; Christmann et al., 2021). The characteristic of a Maxwell material is that it responds elastically on short timescales, while on long timescales viscous creep dominates. This was found already in the 1950's in laboratory studies (Jellinek and Brill, 1956). The solid nature of ice is also the origin
of crack formation, which is key for supraglacial lake drainage and englacial channel formation. On the other hand, the fluid nature of ice supports the closure of englacial channels over time.

In this study, we aim to investigate how the drainage of a massive supraglacial lake at 79NG (see Fig. 1) is facilitated, what role fractures play in the drainage, and if former drainage systems become reactivated. To this end, we use satellite remote sensing data and data from airborne surveys to investigate lake filling, drainage and the englacial pathways of water. To understand
if and how drainage pathways close over time, we incorporate viscoelastic modelling as a case study. We aim to simulate how a drainage channel at the surface of the glacier would change in size over time. To this end, we employ COMice-ve (https://gitlab.awi.de/jchristm/viscoelastic-79ng-greenland), previously used for simulating deformation before (Christmann et al., 2021). The lake we are focussing on is exceptional in size, but most importantly, we selected this one because we were able to build an extensive database from observation over a longer time.

We want to introduce some wording here, as various terms are used in the literature. We do not use the terminology moulin, as our drainage pathways are neither formed by melting nor round in shape, and they are not similar in appearance to the features denoted moulins by Chudley et al. (2019) and Doyle et al. (2013). We are dealing with fractures forming within the lake, but those are not necessarily hydrofractures in the sense that a preexisting fracture became critical by being filled with water.

The manuscript is structured as follows: We first present the methods and give an overview of the extensive database and
the model. Subsequently, we present the results from observations, followed by the simulations. A discussion and a synthesis of data and modelling continue this. As we use data from various sources for several purposes, we provide a comprehensive overview in Tab. 1.





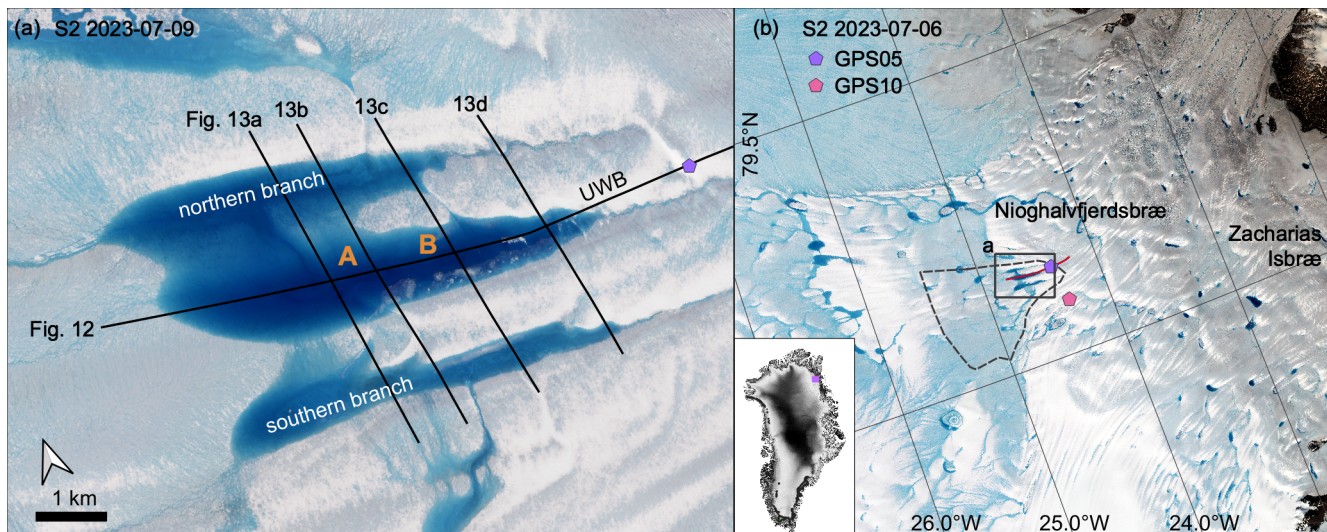

**Figure 1.** Overview figure. Panel a shows the lake on which this study focuses as seen by S2 2023-07-09 with positions of airborne survey crossings as black lines (radar profiles shown in Fig. 12 and 13). Panel b gives a wider overview of the region, superimposed onto a S2 scene from 2023-09-06 is the area for the CARRA (Schyberg et al., 2020) subset shown in Fig. 15 as a dashed line and the GNSS positions presented in Fig. A1 as pink and purple symbols. The inset shows the location of panel b in Greenland.



## 2 Materials and Methods

### 2.1 Optical satellite data

We use optical satellite imagery from different missions: (1) in medium resolution (10 m) S2 (S2) imagery for time series, (2) in high-resolution WorldView2 imagery (1.2 m multichromatic, 0.3 m panchromatic) for snapshots in time to measure the geometry of drainage pathways and to detect the crack locations.

Sentinel-2 (S2) MSIL1C products have been processed using esa's SNAP toolbox. We use bands 2, 3, and 4 to obtain RGB images. The original data was projected onto the resolution of band 2. Most of the imagery we use here is cloud-free, but in 90 some circumstances, we need to use imagery that contains some cloud cover.

We use WorldView imagery from 2019-08-23, 2020-08-19 and 2023-08-01 in the panchromatic mode. The data has not been further processed and is used in the original resolution.

### 2.2 Radar satellite data

In 2006, we use one ALOS-1 single-pol data acquisition to detect fractures formed during lake drainage since, no dual- or 95 quad-pol data was available. ALOS-1 quad-pol data has been used in this study to detect locations of fractures and drainage features for 2007 and 2008. We applied a Pauli-decomposition using the Gamma software package. The resulting product has 20 m pixel resolution and contains single scattering, double scattering and volume scattering as three bands. ALOS-2 data used in this study is only dual-pol data, thus no full Pauli-decomposition could be applied. Here, we are using the different polarisations HH, HV and HH-HV and three bands to identify fractures and drainage features.

We used TerraSAR-X product that was acquisitioned in stripmap mode and enhanced ellipsoid-corrected (EEC). For speckle reduction, we conducted a bi-cubic interpolation to 12.5 m resolution.

### 2.3 Lake volume estimate

In order to estimate the water volume of the supraglacial lake we generated three digital elevation models (DEMs) of the empty lake basin. While two DEMs originate from TanDEM-X data acquired on 2012-05-16 and 2015-11-02 the third DEM 105 was generated from Pleiades data acquired on 2019-05-11. TanDEM-X DEMs were generated using SAR interferometry as described by (Neckel et al., 2013) and the Pleiades DEM was derived by stereo photogrammetry employing NASA's AMES stereo pipeline (Beyer et al., 2018). All DEMs were horizontally and vertically adjusted to the global TanDEM-X DEM (Wessel et al., 2016) in the vicinity of the lake using the pc_align utility provided within the Ames Stereo Pipeline. From the DEMs, we picked shoreline elevations employing the closest available optical satellite scenes of the filled lake. For this, we used an 110 ASTER satellite scene acquired on 2012-07-23, a Landsat-8 scene acquired on 2015-07-24, and two S2 scenes acquired on 2019-07-27 and 2020-07-14. After the lake basin was filled, we calculated the difference of the picked shoreline elevations to the respective DEM of the empty lake. From the integrated differences, we estimated the total water volume of the lake at the four dates of optical satellite acquisitions.





## 2.4 Surface subsidence maps

To estimate surface subsidence in the vicinity of the lake, we will make use of additional Sentinel-1 SAR acquisitions. We employed Sentinel-1 Interferometric Wide (IW) swath Single Look Complex (SLC) data acquired in 2017 and 2019. Interferograms were formed by calculating the phase difference between two Sentinel-1 acquisitions of the same satellite orbit separated by 6 days. The topography-induced phase difference was cancelled with the help of the global TanDEM-X DEM (Wessel et al., 2016). The remaining interferometric phase is mainly attributed to vertical and horizontal surface displacement projected into

the Line Of Sight (LOS) of the satellite (e.g. Rignot et al., 2011; Friedl et al., 2020; Neckel et al., 2021). To cancel the horizontal ice movement, we employed one master interferogram, including data acquisitions from 2019-10-02 and 2019-10-08, respectively. Hence, we attribute the remaining phase difference solely to the vertical surface displacement between August and September 2019 data acquisitions. Depending on the magnitude of the surface displacement and the coherence of the interferograms, unwrapping can be challenging in some cases, requiring different strategies for absolute estimates of surface

displacements. In these cases, we employed speckle-tracking methods on the same data acquisitions as used for interferometry (e.g. Joughin et al., 2016). Finally, the unwrapped interferometric estimates and the speckle-tracking derived fields were projected from the LOS direction to the vertical, representing short-term variations in vertical surface displacement (Fig. 14).

## 2.5 Airborne GNSS

For geo-referencing the optical imagery, laser scanner and radar data, the aircraft position was measured using a dual-frequency

NovAtel OEMV GNSS receiver with a sampling rate of 20 Hz. The flight trajectory was computed using the precise point positioning (PPP) post-processing option, which included precise clocks and ephemerides of the commercial GNSS software package Waypoint 8.90. The accuracy of the post-processed trajectory varies along the track but is better than 0.1 m.

## 2.6 Airborne optical camera

On board AWI's research aircraft, a CANON EOS-1D Mark III Digital Single Lens Reflex (DSLR) camera in combination

with a CANON 14 mm f/2.8L II USM lens is routinely employed. Images were acquired in 6 s intervals and stored with a GNSS time tag in RAW data format. All images were corrected for vignetting effects. During the conversion to JPG format, the original resolution was preserved.

## 2.7 Airborne laser scanner

During the same flight on 2021-07-29, laser scanner data was acquired using a RIEGL LMS-VQ580 with a scan angle of 60°

in about 300 m flight height, this results in a width of the footprint of about 300 m. The mean point-to-point distance is ∼0.5 m. The raw laser data was combined with the post-processed GNSS trajectory and corrected for aircraft altitude and calibration angles, leading to calibrated geo-referenced point cloud (PC) data. We used crossovers to calibrate the system and to derive the elevation accuracy. The geo-referenced PC has an accuracy better than $0.1 \pm 0.1$ m. The bias is $<0.1$ m and varies along the track. It arises from the vertical accuracy of the post-processed GNSS trajectory. The resulting DEM was derived from the





PC by using an inverse distance weighting (IDW) algorithm and a 5 m search radius. Its resolution is 1 m in the horizontal
direction. The elevation was referenced to the EGM2008 geoid Pavlis et al. (2012).

## 2.8 Airborne ice-penetrating radar

Ice penetrating radar is used in this study to obtain the glacier's internal structure. The data was acquired using AWI's Ultra-
WideBand (UWB) Multichannel Coherent Radar Depth Sounder (MCoRDS, version 5), here, with an array of eight antennas
(Hale et al., 2016). The total transmit power is 6 kW. In our survey we used a bandwidth of 370 MHz within the frequency band
of 150 − 520 MHz. The radar was operated with a pulse repetition frequency of 10 kHz and a sampling frequency of 1.6 GHz.
The dynamic range was increased by using alternating sequences of different transmission/recording settings (waveforms).
Short pulses (1 μs) and low receiver gain (11 − 13 dB) were used to image the glacier surface. Longer pulses (3 and 10 μs) with
higher receiver gain (48 dB) are used to image internal features and the ice base. The waveforms are defined, considering the
glacier's ice thickness. Data processing included a pulse compression in range direction and synthetic aperture radar focus-
ing in the along-track direction, which leads to an along-track resolution of 10 m. For the time-to-depth conversion a relative
permittivity of $\varepsilon_r = 3.15$ in ice was used. The survey area is in the ablation zone. Thus, we assume no extensive firn layer to
be present. Therefore, we did not apply any firn correction. after pulse compression, the theoretical range resolution is about
0.23 m. In the last processing stage, the echograms of the two waveforms were concatenated into one radargram, covering the
ice from the surface to the base with a high dynamic range. This has been done for all segments of the flight on 2021-07-29 in
along-flow and across-flow directions.

## 2.9 Copernicus Arctic Regional Reanalysis (CARRA)

To investigate the impact of the atmospheric forcing on the lake formation, we analyzed the summer skin temperatures for June,
July, and August between 1991 and 2022 using the CARRA system (Schyberg et al., 2020). The skin temperature represents
the temperature of the uppermost surface layer that responds instantaneously to changes in surface fluxes. The product of this
analysis consists of daily skin temperatures at 15:00 UTC with a horizontal resolution of 2.5 km. We averaged the daily skin
temperature in the lake's drainage basin and calculated the daily 5-year median temperature with shorter intervals for the first
(1991–1994) and last period (2020–2022).

## 2.10 Modelling

To simulate the evolution of the englacial channel, we chose a two-dimensional top view and modelled the system as a rect-
angular disc with a circular hole. From this point of view, gravity is not taken into account. Thus, the statement of the local
equilibrium reads

$$\operatorname{div} \boldsymbol{\sigma} = 0, \tag{1}$$



**Table 1.** Overview of the data used in this study in respective years. The colour referring to the appearance within the manuscript: x denotes that the data is shown in a figure, x marks that we used this data to extract a feature (e.g. a line, a gully position) which is shown in a figure, and x represents that the data is used to draw information or conclusion that is discussed in the text only. B = blister, E = existence, EC = englacial channels, F = fractures, G = gully, LD = lake drainage, LE = lake extent, LF = lake filling, LV = lake volume, AF = atmospheric forcing

| | Landsat | ASTER | Sentinel 1 | Sentinel 2 | Worldview | Planet | TerraSAR-X | TanDEM-X | ALOS-1/2 | Pleiades | Canon | ALS | UWB | GNSS | CARRA | figure | application |
|---|---|---|---|---|---|---|---|---|---|---|---|---|---|---|---|---|---|
| ≤ 1995 | x | | | | | | | | | | | | | | x | 15 | E, AF |
| 1996-2004 | x | | | | | | | | | | | | | | x | 15 | LF, AF |
| 2005 | x | | | | | | | | | | | | | | x | 3,15 | LE, AF |
| 2006 | x | | | | | | | | x | | | | | | x | 4,15 | LD, F, AF |
| 2007 | | | | | | | | | x | | | | | | x | 4,4,15 | F, AF |
| 2008 | x | | | | | | | | x | | | | | | x | 4,15 | F, AF |
| 2009 | | | | | | | | | | | | | | | x | 4,15 | AF |
| 2010 | | | | | | | | | | | | | | | x | 4,15 | AF |
| 2011 | x | | | | | | | | | | | | | | x | 4,15 | LF, AF |
| 2012 | | x | | | | | | x | | | | | | | x | 3,4,15 | LE, LV, LD, AF |
| 2013 | x | | | | | | | | | | | | | | x | 4,15 | AF |
| 2014 | | | | | | | | | | | | | | | x | 15 | LF, AF |
| 2015 | x | | | | | x | x | | | | | | | | x | 3,4,15 | LV, LD, AF |
| 2016 | | | x | | | x | | | | | | | xx | | x | 3,4,12,,12,15 | F, EC, AF |
| 2017 | | | x | | | | | | | | | | | x | x | 3,4,15,A1 | F, AF |
| 2018 | | | x | | | | | | x | | | | xx | | x | 3,4,12,12,15 | F, EC, AF |
| 2019 | | | x | x | x | x | | | | x | | | | x | x | 3,9,4, 14,15,B1,A1 | F, G, LV, B, AF |
| 2020 | | | xx | x | | | | | | | | | | | x | 3,5,6,4,7,11,15 | F, G, LV, AF |
| 2021 | | | x | | | | | | | | x | x | xx | | x | 7,12,13,12,13,15 | EC, G, AF |
| 2022 | | | xx | | x | | | | x | | | | | | x | 3,6,4,4,7,15 | F, G, LD, AF |
| 2023 | | | xx | x | | | | | | | | | | | x | 1, 3, 8,11,15 | LD, F, G, AF |



with the Cauchy stress tensor $\boldsymbol{\sigma}$. We confine our calculations to small deformations. From the two dimensional displacement vector $\boldsymbol{u} = (u, v)^T$ the linearized strain tensor $\boldsymbol{\varepsilon}$ is derived as

$$\boldsymbol{\varepsilon} = \frac{1}{2}\left(\nabla\boldsymbol{u} + (\nabla\boldsymbol{u})^T\right). \tag{2}$$

For polycrystalline ice, the established rheological model is the Maxwell material, as introduced by Christmann et al. (2021). This material shows instantaneous elastic deformation as well as time-dependent viscous creep. The one-dimensional mechanical equivalent to the Maxwell material is a series of springs and dampers. In the multidimensional case, the volumetric stress is related to reversible elastic volume changes and is, therefore, not time-dependent. We decompose the stress and strain tensors into a volumetric and deviatoric part by performing $(\cdot)^D = (\cdot) - 1/3\mathrm{tr}(\cdot)\boldsymbol{I}$, with the trace operator $\mathrm{tr}(\cdot)$ and the second order identity tensor $\boldsymbol{I}$. The material behaviour can be described by using only the time-dependent deviatoric parts. For the Maxwell material, the total deviatoric strain comprises of an elastic $(\cdot)_\mathrm{e}$ and a viscous $(\cdot)_\mathrm{v}$ part as

$$\boldsymbol{\varepsilon}^D = \boldsymbol{\varepsilon}_\mathrm{v}^D + \boldsymbol{\varepsilon}_\mathrm{e}^D. \tag{3}$$

The deviatoric stress in the viscous element with viscosity $\eta$ equals the deviatoric stress in the elastic element with shear modulus $\mu$ and reads

$$\boldsymbol{\sigma}^D = 2\eta\dot{\boldsymbol{\varepsilon}}_\mathrm{v}^D = 2\mu\boldsymbol{\varepsilon}_\mathrm{e}^D, \tag{4}$$

with the time derivative $\dot{(\cdot)} = \mathrm{d}(\cdot)/\mathrm{d}t$. Inserting eq. (3) into eq. (4) we obtain the evolution law for the deviatoric strain

$$\mu(\boldsymbol{\varepsilon}^D - \boldsymbol{\varepsilon}_\mathrm{v}^D) = \eta\dot{\boldsymbol{\varepsilon}}_\mathrm{v}^D. \tag{5}$$

The resulting stress to fulfill the equilibrium statement (1) follows from the material law of the elastic element as

$$\boldsymbol{\sigma} = K\,\mathrm{tr}(\boldsymbol{\varepsilon})\boldsymbol{I} + 2\mu\boldsymbol{\varepsilon}_\mathrm{e}^D, \tag{6}$$

with the bulk modulus $K = E\mu/(3(3\mu - E))$, where $E$ is the elastic Young's modulus.

The system of equations is solved with the Finite-Element software COMSOL MULTIPHYSICS 5.4. For the simulation, we chose typical values for polycrystalline ice such as

$$E = 9\,\mathrm{GPa}$$

$$\nu = 0.325$$

$$\mu = 3.396\,\mathrm{GPa}$$

$$\eta = 3 \cdot 10^{14}\,\mathrm{Pa\,s}$$

$$p_0 = 1 \cdot 10^5\,\mathrm{Pa} \text{ for } 0 < t < 10\,\mathrm{d}.$$



The edge length of the rectangular disc holds $20\,\text{km}$ and the radius of the hole is chosen to be $20\,\text{m}$ (Fig. 2). A horizontal displacement is applied for every time step $t$ on the boundaries to model the accelerated ice flow. As the left boundary condition, we chose $3.8\,\text{m/d} \cdot t$ in accordance with remote sensing-based velocities and on the right boundary $3.9\,\text{m/d} \cdot t$. The upper and lower boundary are fixed in the vertical direction but can move freely in the horizontal direction. The water pressure inside the

205   channel before and after drainage is modelled by applying traction in the radial direction. The traction is applied for $10\,\text{d}$ and then set to zero for another $10\,\text{d}$.

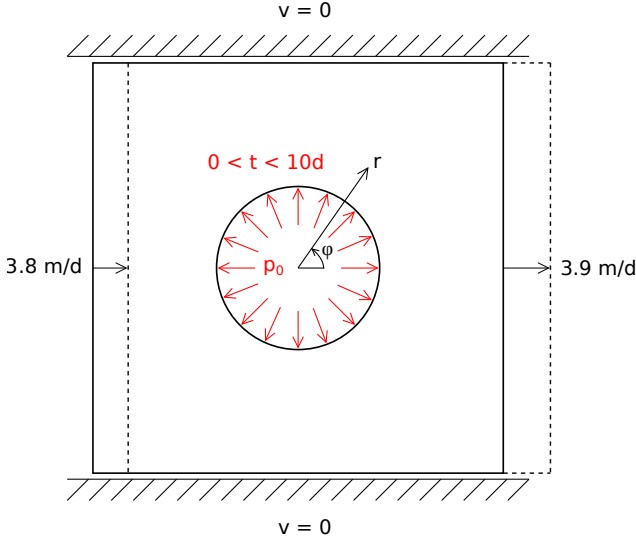

**Figure 2.** Coordinate system and boundary conditions for disc and hole.



# 3 Results

## 3.1 Observations

In the following, we first present the results from the satellite imagery, which is the most extensive part of this section. Subse-
210 quently, we will show observations from the airborne survey, and lastly, we will provide data on atmospheric forcing.

All data available to us in the Landsat, ASTER and Sentinel archives were screened. In addition, we used ALOS-1 and ALOS-
2 data to identify fractures and gullies. Although the data coverage in the 1980's and 1990's is sparse, one can constrain the
initiation of this lake to the year 1995. It took about ten years until the first drainage occurred (see Tab. 2). Figure 3 presents
the lake extent prior to drainage.

**Table 2.** Timing of start and completion of filling and drainage.

| begin filling | filling complete | drainage start | duration (d) | volume (m$^3$) |
|---|---|---|---|---|
| 1995 | Aug 2005 | < 2006-06-03 | | |
| 2012-07-10 | 2012-07-22 | $\geq$2012-07-23 | $\leq$2 | $0.65 \cdot 10^8$ |
| 2013-07-24 | 2014-07-24 | $\geq$2015-07-24 | $\leq$3 | $1.23 \cdot 10^8$ |
| 2019-07-12 | 2019-07-26 | 2019-07-30 | 2-13 | $0.72 \cdot 10^8$ |
| 2020-07-07 | 2020-07-14 | $\geq$2020-07-15 | $\leq$3 | $0.49 \cdot 10^8$ |
| 2022-07-20 | 2022-07-31 | $\geq$2022-08-15 | 4 | |
| 2023-07-04 | 2023-07-11 | 2023-07-11 | 1 | |

We digitised lake outlines before the drainage event (shown in Fig. 3a) based on the last available imagery before major
drainage: 2005-07-20 (Landsat), 2012-07-25 (ASTER), 2015-07-24 (TSX), 2019-07-26 (S2), 2021-08-13 (S2), 2022-08-13
(S2) and 2023-07-09 (S2). Below, the details for the individual years of drainage are presented.

The first drainage took place between the fall of 2005 and June 2006. The lake area before the drainage was the largest on
record. The lake surface was no longer a flat snow-covered area in a Landsat image acquired on 2006-06-03 (not shown). This
was the first drainage event at this lake during the satellite observational period. At this time, the surface structure was similar
to what ALOS-1 L-band imagery from 2007-04-03 (Fig. 4), almost a year later, revealed.



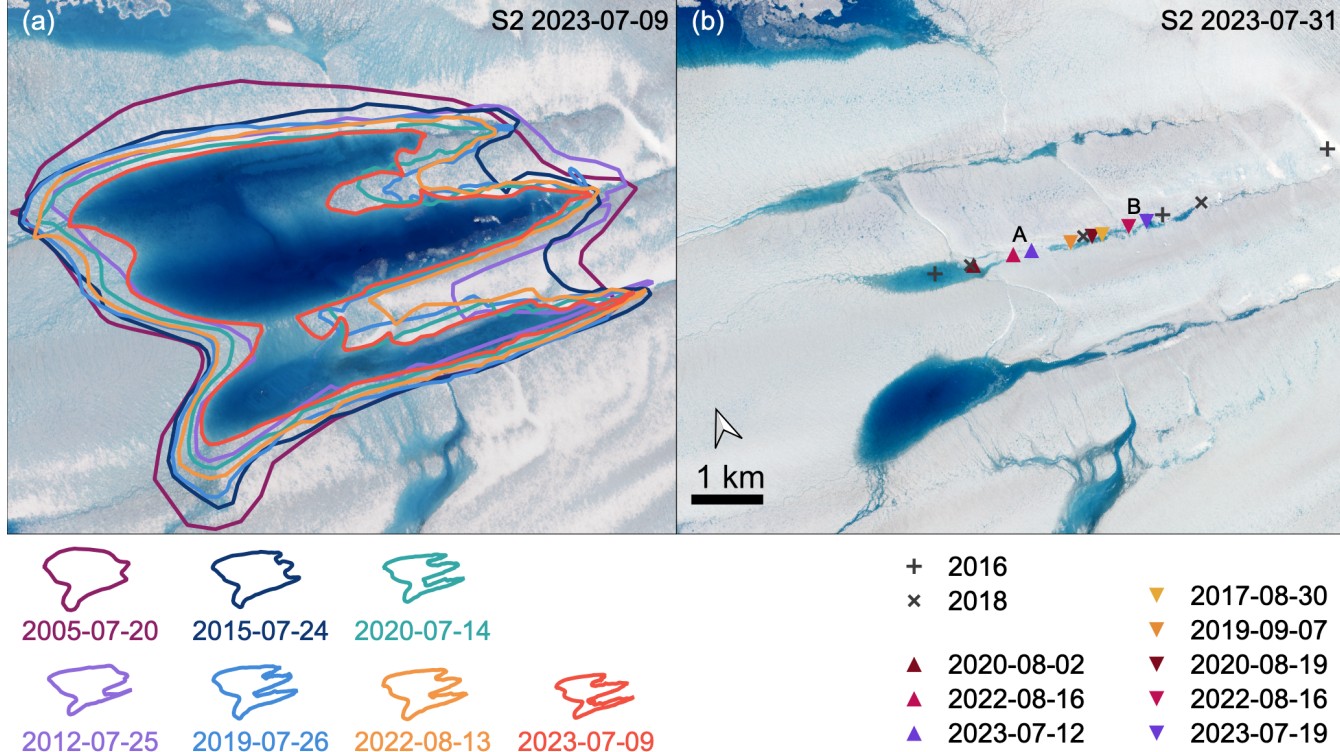

**Figure 3.** a Maximum lake extent before lake drainage for all drainage events (coloured lines) superimposed over S2 imagery from 2023-07-09; b Gully locations in areas A and B shown as triangles. Colour corresponds to the year of gully formation. Grey crosses mark locations with englacial reflections in the UWB radargrams (Fig. 12) in the years 2016 and 2018.

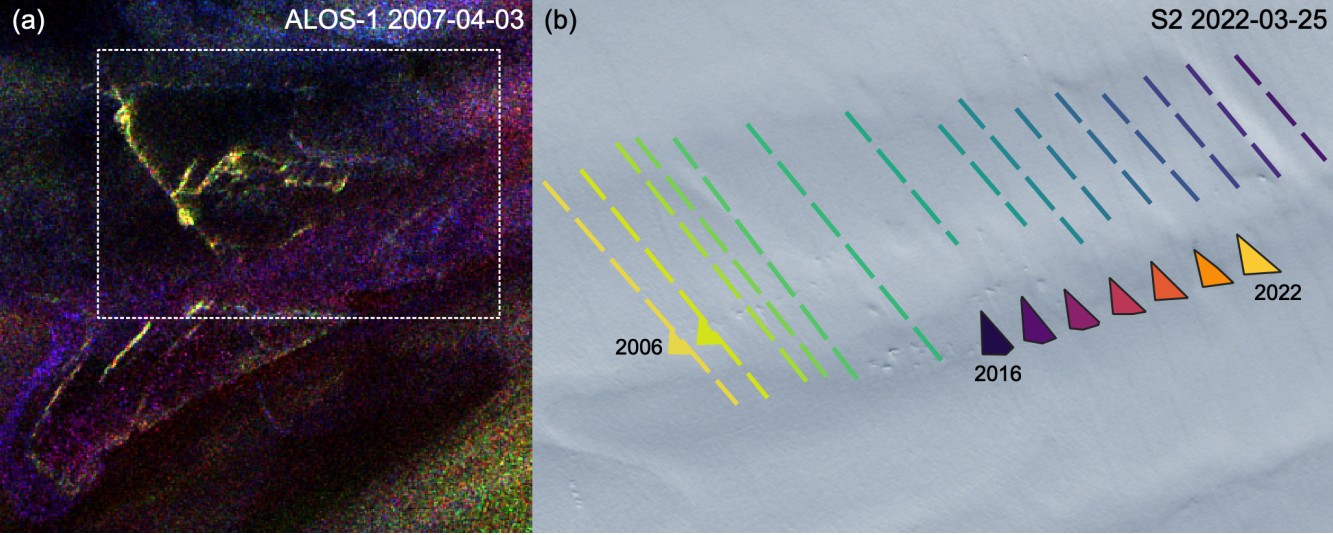

**Figure 4.** Panel a displays a Pauli decomposition from an ALOS-1 quad-pol acquisition on 2007-04-03. The right panel b displays a S2 image 2022-03-25 in the background. Coloured lines denote the fracture from the 2005/06 drainage event and its evolution over time. Note the shade of the background image of the last grey dashed line, which indicates a topography step. Coloured triangles mark the triangular-shaped feature from the 2015 drainage event that remains visible in subsequent years in optical imagery, although with a fading appearance.



The second drainage occurred in 2012 between 2012-07-22 and 2012-07-26 (ASTER). The lake extent before drainage was smaller than in 2005 (Fig. 3a).

In 2013, the middle branch of the lake filled, but no drainage occurred, indicating that the drainage pathways from 2012 were shut. At the end of August, the lake's surface froze over. In the summer of 2014, the lake filled, but no drainage took place. The lake's surface froze over at the end of summer 2013 and 2014.

In 2015, the third drainage took place between 2015-07-24 (Fig. 3a) and 2015-07-27 (Landsat, not shown). The drainage
occurred while the lake was still covered with ice, preventing any gully analysis. However, together with the drainage, cracks have been formed (also well visible in a TSX acquisition on 2015-08-04, not shown), which are discussed in the section below. End of August (2015-08-26, not shown), an overflow from a northern lake is found in TSX imagery.

In 2016, the lake did not fill entirely, although some melting occurred (visible in S2 imagery, which we do not display here). This is consistent with a still open and active drainage system. TSX imagery 2016-07-21 (not shown) exhibits open water in
the northern, central, and southern branches. In the central branch, the shape of the ponding is comparable to other years but has an ending in the form of a 155 m long line. On 2016-09-09, a hole was visible at the southern end of that line. It is the first feature that we identify as gully. It is located in area B (see Fig. 1b) and shown in Fig. 3b. In S2 imagery, surface cracks and drainage features are visible, which will be discussed below.

In 2017, melting started late and filled the lake from 2017-07-26 with little amount of water. The peak was reached on 2017-08-01 (both not shown here). GNSS data shows velocity change between 2017-07-31 and 2017-08-06 of about 10%, indicating higher lubrication of the bed that can be expected to originate from the lake (Appendix A), as the S2 scene on 2017-08-09 shows already a decline in the water body. This points towards a still active, or easy to reactivate, drainage system. The gully in area B has moved with the speed of the glacier (yellow mark for 2017-08-30 in Fig. 3b).

The year 2018 had a rather cold summer in which no substantial melting occurred at the lake's elevation. A nearby GNSS station was covered by snow the entire summer. S2 scenes do not show any meltwater, and ALOS-2 data does not exhibit any new signs of fracture.

The fourth drainage event happened in 2019: The largest lake extent in 2019 was found on 2019-07-26 (Fig. 3a), again smaller than in 2005, 2012 and 2015. S2 acquisitions cover the drainage. While the lake was still filled on 2019-07-30 (not shown), the scene from 2019-07-31 (not shown) shows that the lake volume reduced drastically, but still, deeper areas were filled. One drainage location can be identified: a gully at location A can, however, be only inferred due to ponding end of summer, 2019-08-22 (S2), which has a form that is consistent with a drainage pathway at this location. Also, in WorldView on 2018-
08-23, this location is the end of a water body, as shown in Fig. 10. It looks as if the water has been flowing into a triangular area with 54, 44 and 67 m edge lengths formed by cracks, and it appears visually deeper than the surroundings. Moreover, a S2 scene from 2019-08-22 shows a concentration of cracks in location B (Fig. 3b). This image also reveals a crack 1060 m



downstream from Gully A in that year, which was not the main drainage location in 2019. This crack could be a remnant of an older drainage. With an average ice flow speed of 285 m/a, this drainage pathway could have been formed in the 2015 drainage

event at roughly the same location as the gully in region A in 2019.

Drainage event number five occurred in 2020, the first one that took place just one year after a prior drainage event. The lake was filled on 2020-07-14 (Fig. 6a), just seven days after it started to fill. Only one day later on 2020-07-15, a large amount was drained, but drainage continued until 2020-07-17 it is mainly drained (the southern branch still contains a significant

amount of water). A gully in the 10 m S2 resolution imagery is most striking. On 2020-08-19 a high-resolution scene (WV) was acquired, displayed in Fig. 5. At the location of the gully visible a month ago in S2 imagery, a triangular-shaped area is filled with water (230, 213, 170 m side lengths), and a heap of ice blocks is found (largest block measures 6 m in width). Outside this gully, southeast of the heap, larger ice blocks are found (the closest is 318 m away from the gully). The largest one measures 20 m in width, and the majority is in the order of 10 m in width. At the end of summer, the central branch of

the lake is drained entirely, and the southern branch is almost empty (Fig. 6b). Only the northern branch contains some more water, which appears to come from an overflow of the neighbouring lake.

The meltwater in the summer of 2021 did not lead to a complete refilling, and besides some water ponds in the middle branch, no drainage took place by the end of summer. The gully from 2020 was still active. In 2021 we surveyed the lake with the

UWB, ALS, and Canon camera on 2021-07-29 (Fig. 7c,d). The optical images from the onboard camera prove that the gully still has its horizontal geometry of 2020-08-19. The gully appears in optical images to be covered with snow over most of its area except one corner. Laser scanner data reveals this corner to be 16 m deeper than the surrounding surface. Note that this is not the true depth of the gully, as the flight elevation and speed do not allow for a sounding of the depth. Just one day later (2021-07-30), S2 imagery shows that the lake ice is entirely melted and the width of the ponding area at area A is 278 m. By

2021-08-08 this width is reduced to 96 m at the same location. The gully at location A-2020 has moved between 2020-08-19 and 2021-09-02 for 297 m, which is consistent with a glacier velocity of 290 m/a. At location B, the gully is still 'blocked' with ice fragments and is elevated about 1 m higher than the surrounding lake area, underlining the finding that ice fragments are piling up.



**Figure 5.** Gully in the drainage event in 2020. The high-resolution image is a multichromatic WorldView image acquired on 2020-08-19. Panel a displays the central branch of the lake and indicates the location of the two areas of panel b, c.

In 2022, before the lake drainage, the gully at location A was visible in the filled lake in S2 imagery for several days. At
location B, we identified some floating ice blocks near the gully on 2022-08-15 (Fig. 6c, the scene is unfortunately cloudy).





A day later, the lake drained substantially. Ice blocks are visible next to the gully at location B. High-resolution imagery from 2022-08-20 (4–5 d after drainage) reveals that the size of the gully in area A is about the same as in 2020 (location marked with a purple mark in Fig. 3b). A change in the size of the gully in the region B cannot be estimated, as the gully was covered with ice fragments and water in 2020. The ice blocks formed before drainage appear to have a size of 8–20 m. The closest is only 10 m away from the gully, and the farthest distance (in the coverage of this acquisition) is about 1.2 km. After the drainage in 2022, the lake still contains water, as seen in the optical satellite imagery (Fig. 6d). As the gully has moved with the glacier, it is now at a slightly higher elevation location than in the years before, preventing a complete discharge of water.






**Figure 6.** Before and after drainage: (a, c) lake filling last day before drainage; (b,d) last day of the season after drainage.



**Figure 7.** Panel a and b are showing the gully at location A (Fig. 3) in high resolution. Panel a is a pan-chromatic image, and panel b displays a multi-chromatic image. The time between the two high-resolution images is two years, and the geometry of the gully remains the same. In panel c the topography of the surface is shown, while panel d shows synchronous optical imagery of the onboard camera.



In 2023, the lake has filled rapidly. Over only 7 days, the lake filled to an extent slightly less than in 2007 and significantly less than in 2020 and 2022 (Fig. 3a). Nevertheless, between 2023-07-11 and 2023-07-12, most of the lake drained, with some filling remaining in the southern branch. This is the earliest recorded drainage so far. No new gullies exist, but the two from 2022 can be identified even in S2 imagery (Fig. 3b). This indicates a reactivation of the gullies.

We find that the blocks from 2022 are before and after the drainage in 2023 still in the same constellation, indicating that they did not become afloat and no drift took place. We envisage the following reason for that: the water level at the end of 2022 was low enough that it froze entirely during winter, encapturing the blocks. This may also have led to a more rapid filling of the lake, as less water volume is needed to reach the same areal extent than in other years.

Figure 8 displays the situation on 2023-08-01 with water still flowing into cracks. We will discuss the details of the fracture formation and reactivation further below.







**Figure 8.** Detailed view of the 2023 drainage pathways. Panel a shows a WorldView multi-chromatic image from 2023-08-1 in the background, with black lines superimposed that denote crack locations. Fractures that existed already in 2022 are marked as bold lines, and new cracks formed in 2023 as thin lines. The crosses denote principal directions retrieved from inverse modelling. Panels b-d display panchromatic imagery from the same acquisition as in panel a at locations marked in panel a. Dark colour represents water.



**Crack formation associated with drainage**

Next, we present evidence of crack formation in accordance with lake drainage. As the drainage events in 2005/06 and 2015
are different in type from the events in 2019 onwards, we start with these two events. In the events 2005/06 and 2015, a long
crack transverse to the ice flow direction appears over the entire width of the lake. A triangular-shaped feature is visible in both
cases, with one side of the triangle located on the main, long crack. Pauli decomposition of the L-band ALOS-1 and ALOS-2
data reveals that the reflection is a double bounce, thus either from a crack face or a face of a higher elevated block of ice. The
triangular-shaped features also appear as double-bounce features.

ALOS-1 quad-pol imagery from 2007-04-03 (Fig. 4a) reveals strong double bounce reflectors. Among those reflectors is a
linear feature with a triangular-shaped area detected as a line and a triangle in Fig. 4b. The same feature is also visible in a
single-pol acquisition from 2006-07-15, and the distance equals the glacier motion. Its sides have a length of 200–300 m. No
double-bounce reflectors are visible in an ALOS-1 quad-pol coverage on 2008-11-04 and 2010-12-14.

The long crack in the 2005/06 event is shown in Fig.4b. This crack can be followed over several years, although the appear-
ance changes. Its position over time is shown as coloured, dashed lines. The distance between the lines is not equal because
satellite imagery is not available at the same time each year. Note that even in the background image from 2022, the old crack
is visible as a shade. In optical imagery, the long crack causes significant shades, indicating a vertical displacement of both
crack faces, with the downstream side being elevated higher than the upstream side. In addition, the filling of the lake reflects
in years after 2006 a blocking took place, thus a part of this crack marks the lake margin. Unfortunately, our database does not
allow us to analyse if similar features also appeared in the 2012 drainage event.

In Fig. 4b, we present with coloured dashed lines the fracture of the 2015 drainage and as coloured areas the triangular-
shaped feature tracked over the years. In that drainage event, the crack is 2725 m long and the triangular feature has a side
length of 250–300 m. Both features are still well visible in ALOS-2 imagery from 2016-12-18 as surface reflectors, which is
17 months after their formation. The high-resolution WV imagery from 2020 also shows a triangular-shaped feature of similar
size (170–230 m side length), although strongly eroded, and it is still identifiable in 2022 (Fig. 4). On the other hand, the crack
from 2015 can not be identified in 2021. The location of the triangular-shaped feature is slightly off the trajectory between
regions A–B, 200–500 m further south. Most importantly, its location is higher elevated, at the margin of the lake's middle
branch, and is thus unlikely the main drainage channel.

We can identify the cracks associated with the drainage event based on the high-resolution imagery. Figure 8 shows the
situation in 2023 and Fig. 9 shows an overview of 2019, 2020, 2022 and 2023. We find cracks in all three branches of the lake,
but the densest network of cracks is situated in the middle branch, while the southern branch has only a few cracks in all the
years. The crack system is centred around the gullies. From there, cracks have reached in some years into the northern (2019,
2020, 2023) and southern branches (2019, 2022).



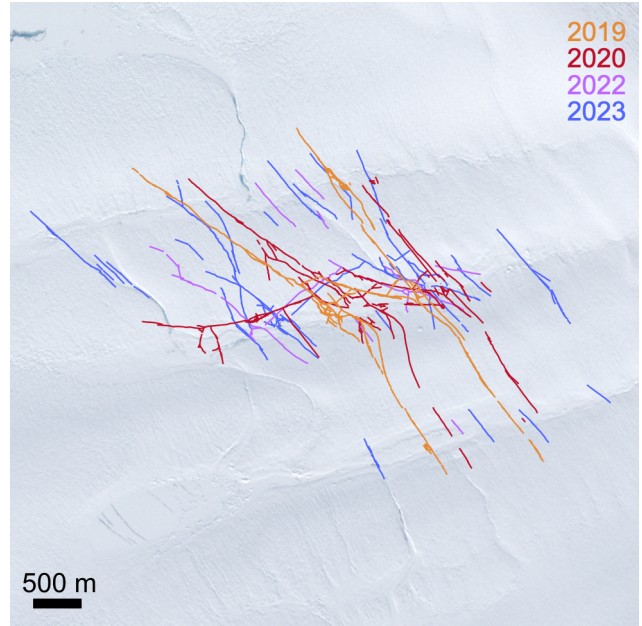

**Figure 9.** Cracks detected in high-resolution optical imagery in 2019, 2020, 2022 and 2023. Background image S2 2023-09-05.

Wherever cracks are crossing river systems inside the lake area, it appears that water is flowing into the cracks (Fig. 8c, d)
and this even weeks after the main drainage event occurred.

The crosses in Fig. 8 denote the principal stress directions based on inverse modelling (similar to Humbert et al. (2023a). These directions match extremely well with many crack orientations, indicating tensile mode (mode I) fracture as the dominant fracture mode. Close inspection of the surface of the lake ground shows narrow and potentially shallow cracks to exist in the area inside and outside the lake, as can be seen in Fig. 8d. Those match perfectly with the principal directions. However, crack
orientations that are not aligned with the principle directions exist, too. One example is visible in Fig. 8d. The main crack formed with the drainage deviates from the direction of the shallow, narrow cracks that resemble a background crack pattern. Similar to Humbert et al. (2023a), we find crack tips being accompanied by tiny cracks in 45°, which are formed in the main shear direction. One of these examples can be found in Fig. 8b. Once these cracks propagate at 45°, they form a connection from one row of narrow cracks to the next, where the crack propagation follows the main principle direction for a while until
it again jumps onto the 45° and so on. This type of crack propagation is responsible for cracks deviating from the principal directions.

As the sun has been low at one acquisition, we can use shadows to assess whether an uplift occurred across a rift. Figure 10 shows an inset around a gully with an arrow pointing to a shadow. We infer that along this crack, an uplift has occurred and still exists 23 d after the drainage began. The conversion of shadow length to object height reveals the step between the crack
faces to be roughly between 0.8–1.0 m.



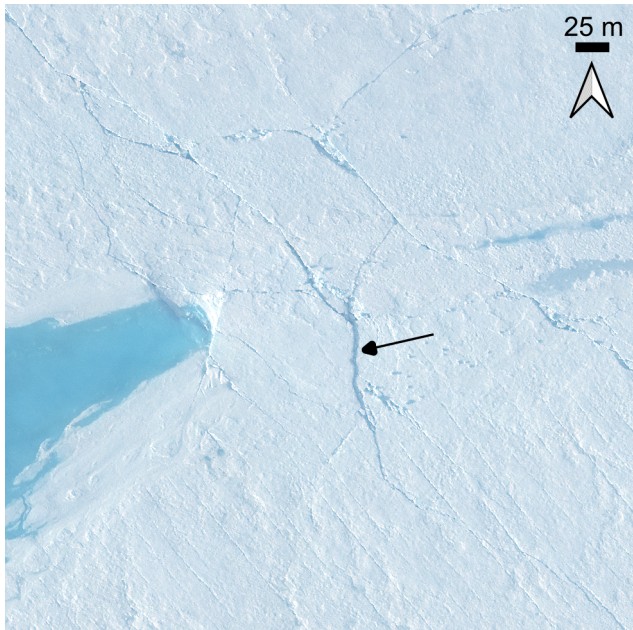

**Figure 10.** Gully and cracks with uplift (WorldView 2019-08-23). The arrow points towards shadows that mark the uplift of the ice between the gully and this crack.

**Gully refilling**

We observed a refilling of the gully at location B in the years 2020 and 2023. Figure 11 displays the satellite imagery in 2023 and 2020 in the upper and lower rows, respectively. After the main drainage event, we find an open gully, where the darker pixels are shadows, and the brighter pixels are reflections of sunlight. On 2023-07-31, the gully is filled with water, and a patch around the gully is filled, too. In 2020, the scenario we observed was slightly different: within four days, the gully overflowed. Some ice blocks were visible, and another four days later, it was empty again. Sixteen days after that, the gully overflowed, and the area filled with water remains until it freezes over. On 2020-07-28, the lake ground appeared wetter than the days before and after. This is not the case in 2023. In both years, cloud cover prevents a closer inspection, thus both the timeline and indications of the origin of the water need to be treated with care.





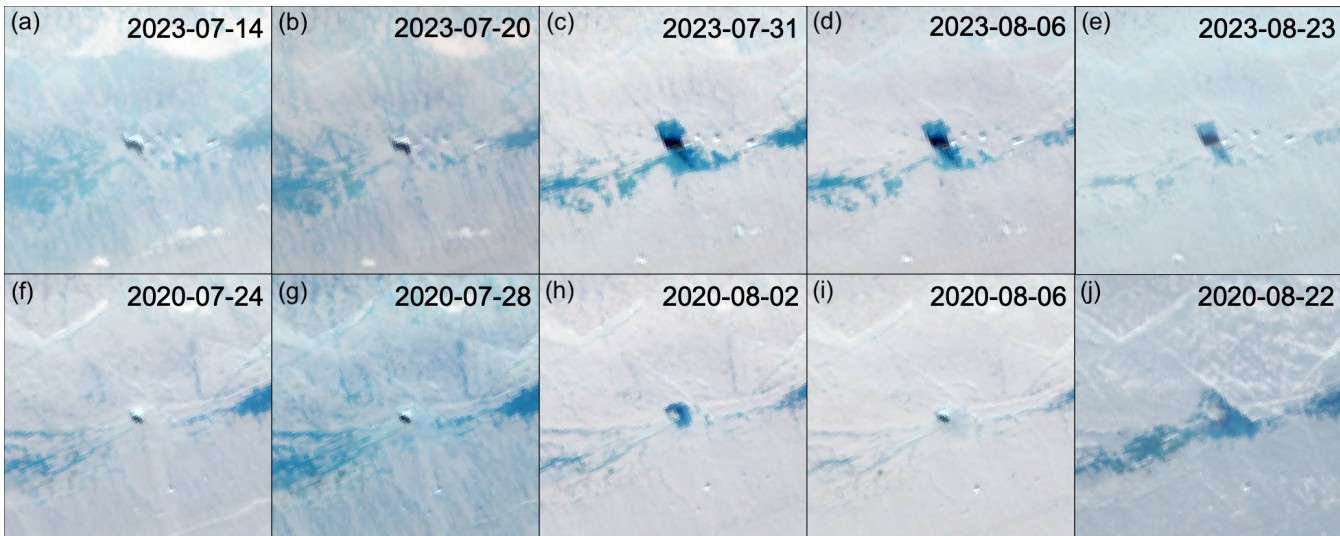

**Figure 11.** Refilling of the gully after drainage. The upper row (a-e) shows the temporal evolution in 2023 and the lower row (f-j) in 2020.

**Englacial channels**

Figure 12 presents three radargrams that are all acquired at the same flowline across the lake, but in different years: 2016-06-27 (Fig. 12a), 2018-04-11 (Fig. 12b), and 2021-07-29 (Fig. 12c). Below the lake, the radargrams exhibit englacial reflections that are quite distinct from normal stratigraphic internal layering or folds near the base. Downstream of the presented profile, no englacial reflections are found near the lake. Please note that with our flight profile, we are likely crossing a three-dimensional

feature, while we do not have any information in the perpendicular direction.

We find in 2016 three prominent features 200–300 m above the ice base (at distances 4.5, 7.8, 10.5 km) (denoted F1, F2, F3 in Fig.12a). There are further features visible that appear as thin, oblique, or wing-shaped reflections with lower amplitude (at distances 3 to 7.5 km) of which the one closest to the ice surface (lake ground) is roughly at 500 m depth (W1 in Fig.12a)). In 2018, all three of the prominent features recorded in 2016 are still present. Feature F1 and F2 have changed their width and

strength; they have become narrower and have less amplitude than 2016. One new prominent feature appeared at a distance of about 6.8 km (denoted F4 in Fig.12b) and is located about 300 m above the bed.

In the radargram from 2021, Fig.12c), the uppermost englacial feature is only 300 m below the lake ground. It is the only profile containing reflections high up in the ice column. At 5.5–7.5 km distance, a wide feature is visible, which consists of several individual hyperbolas (denoted F5 in Fig.12c). Above and below we find two smeared/diffuse reflections: one from

about 250 m below the lake ground pointing towards feature F5 and in the lower 250 m oriented towards the base.

One of the less strong reflections (W1) from 2016 can be found with a strikingly similar shape in the radargram from 2018 and 2021, located at a position approx. 1 km distance downstream, which is consistent with the surface velocity of the glacier (approx. 300 m/yr). A triplet of features close to the ice base in the radargram of 2018 (T1) is visible in the 2021 radargram in the same shape, just in approx. 1 km distance downstream.



The feature farthest downstream (to the right (F3 in Fig. 12a)) can be tracked over the five years. As in the other cases, its travel distance corresponds to the glacier's velocity. Interestingly, in 2021, it is smeared out, which could potentially originate from vertical shear while climbing across the bedrock step. Using the glacier velocity to trace this feature upstream shows that it would have been located in 2005 at location A. Thus, it may be the remnants of the first drainage event.

     In all three radargrams, we find stripes of higher reflection power near the bed, which are almost horizontal, while the bed is
sloped at a different angle. These may arise from side reflections at the ice base, which may arise from bedrock topography or subglacial channels.



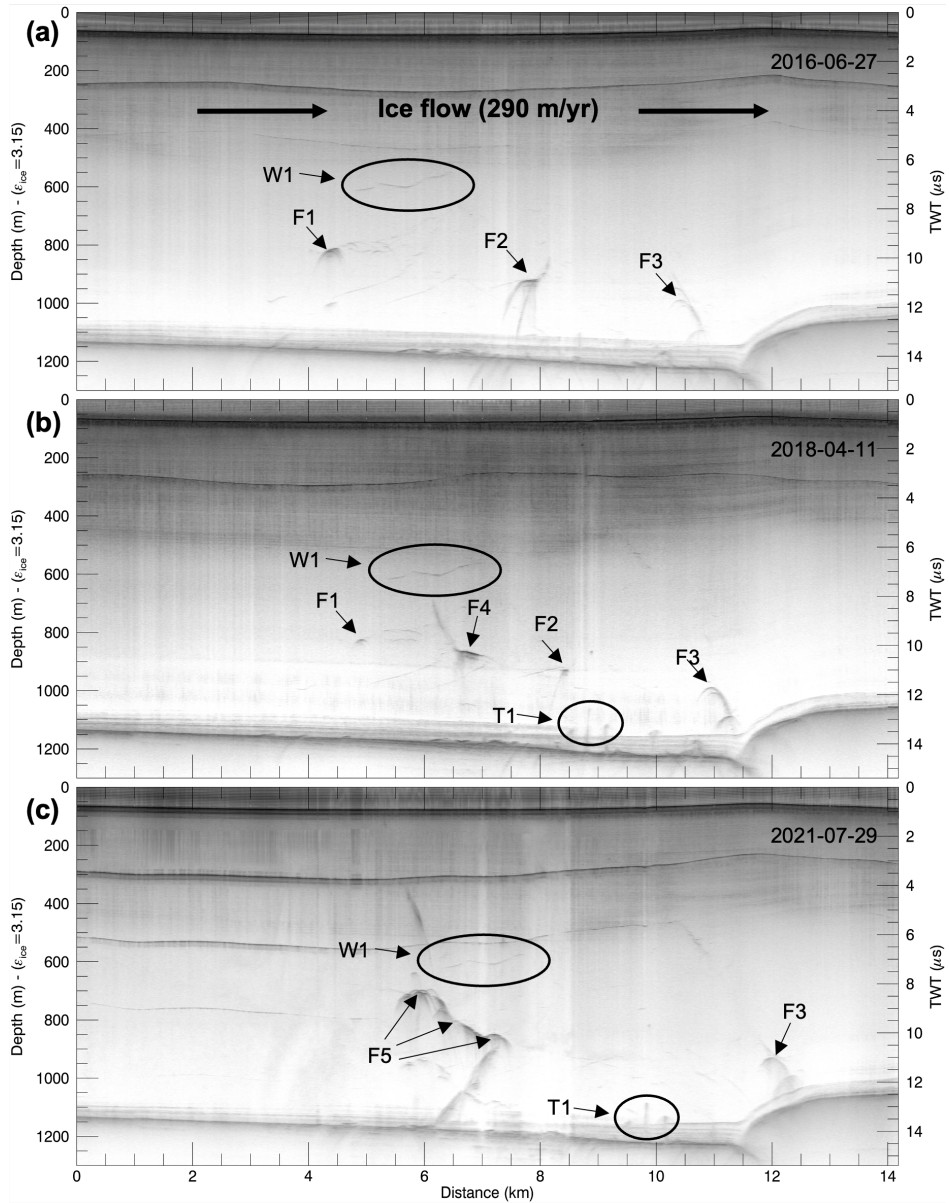

**Figure 12.** Radargrams along the cross section in along-flow direction of the glacier (red line in Fig.1) acquired 2016 (panel a), 2018 (panel b) and 2021 (panel c). All three radargrams show the same cross-section. The flow direction of the glacier is from left to right. The location of the transects is shown in Fig. 1.



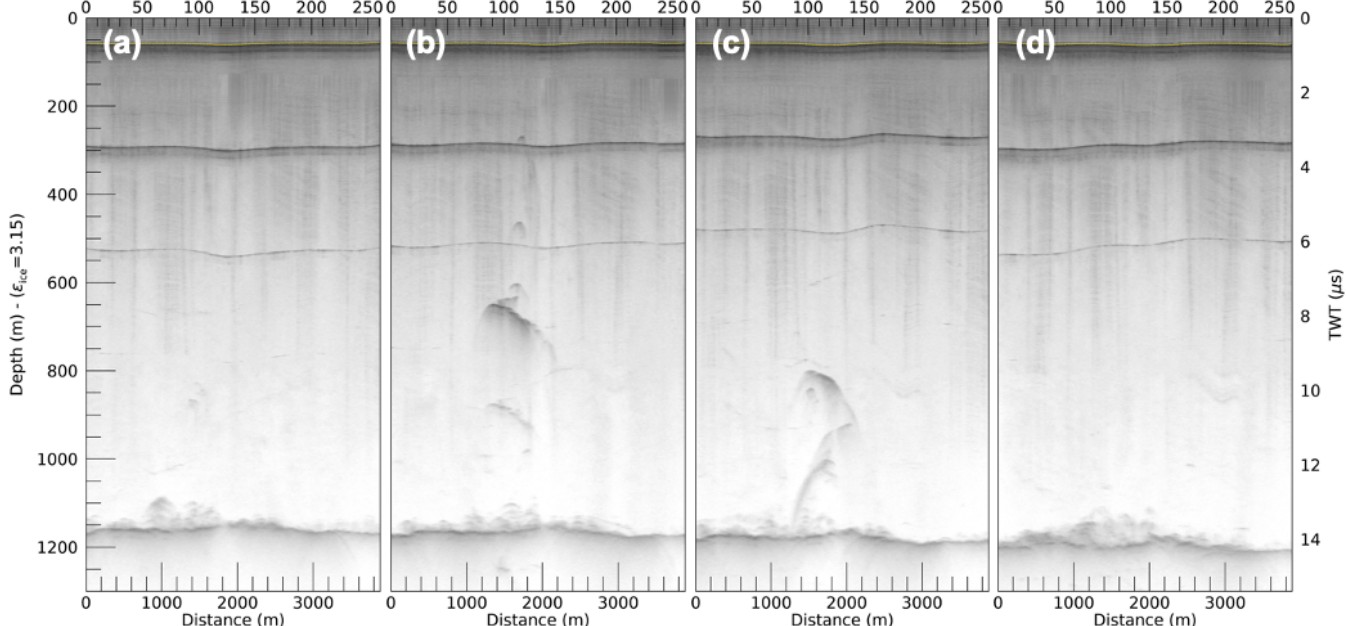

**Figure 13.** Radargramms in across-flow direction from the upstream end of the lake to the downstream end. The location of the transects is shown in Fig. 1.

In 2021, we acquired four transects across the flow direction in addition to the along-flow direction. Their position is shown in Fig. 1a. The one furthest upstream is acquired about 280 m upstream of the gully in region A. It exhibits no englacial features (Fig. 13a). The second one is recorded 380 m downstream of the gully at location A. This radargram contains strong englacial
reflections, which are located about 500 m above the base (Fig. 13b). The lateral extent of these englacial features is 1 km. Further up in the ice column, a narrow feature is found, which is located 400 m below the ice surface/lake ground. Very close to the base, a number of hyperbolas are visible. The third across-flow profile is located just 150 m upstream of the gully at location B from the 2020 event. We find again a strong internal reflector, situated about 300–400 m above the base and a lateral dimension of 1 km (Fig. 13c). Below this reflector, we also find hyperbolas close to the base. The fourth cross-section within
the lake area is 1.2 km downstream of the gully at B. This radargram exhibits no strong englacial feature but has a wider and higher area of hyperbolas close to the base (Fig. 13d).



## Blister evolution



**Figure 14.** Surface elevation change based on interferograms with red colour representing subsidence. The four panels cover a period from 3-4 to 7-8 weeks after the main drainage. The dashed white line represents the maximum filling on 2019-07-26, while the white, bold line represents the cracks on 2019-08-23. The background image is a S2 image from 2019-09-07.



We use double-differential interferometry data from 2019 to assess the evolution of the surface elevation after the drainage event. Unfortunately, the surface condition allows this only three weeks after the drainage onwards. Figure 14 displays the displacement from 3–4 to 7–8 weeks after the main drainage and Fig. B1 the interferograms. The centre of the fringes is at the beginning slightly upstream of the gully location. The largest displacement occurred directly in the lake. Still, it reaches about 2.3 km in downstream direction, 2.0 km in upstream direction and even ∼6 km in northward direction (going beyond the neighbouring lake). The downstream end of the fringes coincides with the bedrock step visible in Fig. 12. In the subsequent weeks, this extent reduces, and the centre of the fringes moves slightly downstream to the gully. The magnitude outside the lake is significantly lower, and in the order of magnitude, that is consistent with vertical displacement detected by the GNSS stations (see Fig. A1). During the time covered by the interferogram, the level of the remaining water patches does not change anymore, but lake ice is formed, which is not covered with a snow layer (last optical imagery from 2019-09-07). This indicates that the elevation change is due to a blister formed during the drainage underneath the lake.

**Atmospheric forcing**

We present skin temperatures in the catchment of the lake from the 1990s to 2022 in Fig. 15. Each year is shown as a grey line, while 5-yr median skin temperatures are superimposed in colour (the first and last average contain fewer years). While the median shows little variation in magnitude in mid-July, early and late summer temperatures have increased substantially. This is increasing the period in which melt occurs and, as a consequence, the amount of melt water available. The period 2010-2014 is the one with the longest time of the 5-yr median skin temperature above melting point. The first half of the 1990s is the shortest melt period, while the second half marks the onset of an increase in the melt period in this record. As we computed here, the area of the catchment can serve as an indicator of the formation and filling of the lake. Below, we will synthesise this dataset with lake filling and drainage observations.





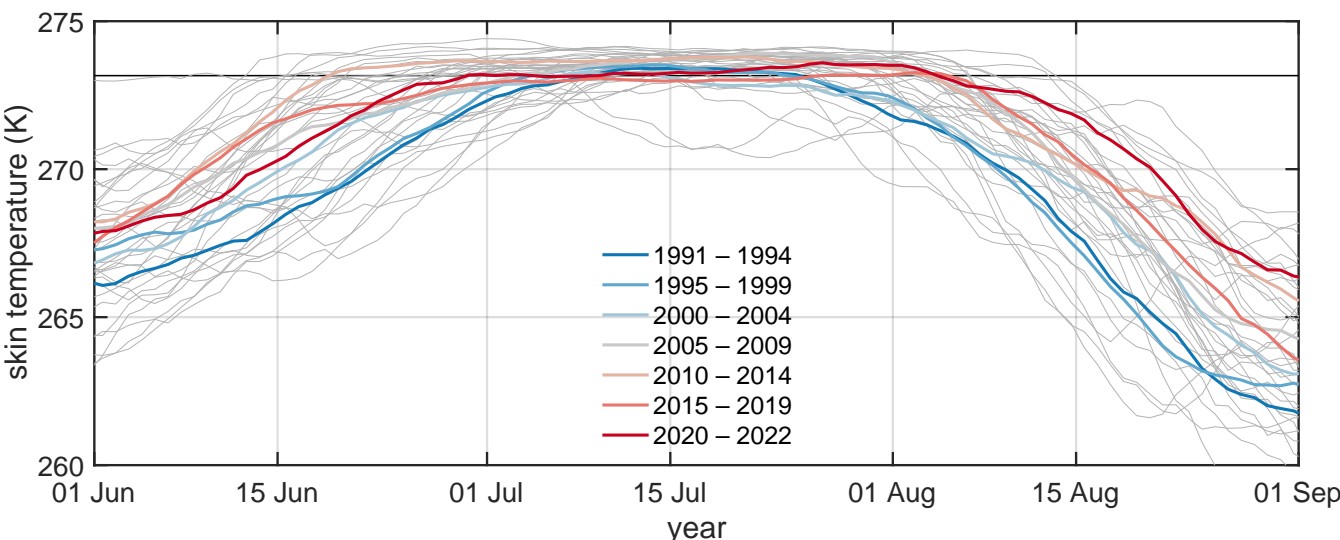

**Figure 15.** Atmospheric forcing: CARRA skin temperatures 1990's to present. Each year is displayed in grey, whereas the median temperatures of five years (the first and last time periods are shorter) are superimposed in colour.

### 3.2 Modelling

In order to evaluate the shape and size stability of englacial channels the model in Sect. 2.10 was set up. The deformation for
a time span of 10 d or 20 d is simulated. Figure 16 shows the resulting radial displacement on the hole boundary relative to the
rigid body movement of $3.8\,\mathrm{m/d}$ for three-time steps. The shape of the curves $t > 0$ indicates that the initially circular channel
was deformed into an elliptical shape with a mean change of radius of $0.032\,\mathrm{m}$ for $t = 20\,\mathrm{d}$. The corresponding surface change
is $0.3\%$. From this, we conclude that our simulation predicts no significant surface change of the channel but a slight widening.

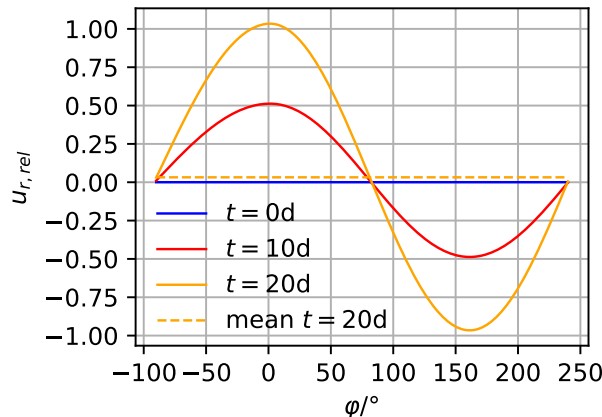

**Figure 16.** Simulated radial displacement of hole boundary relative to rigid body movement.





## 4 Discussion

We find crack formation with lake drainage near the gully at location B. We envisage the drainage to occur not only through the gully but also through the cracks. In the optical imagery after drainage, the cracks are very narrow, and we assume that they closed (not healed) elastically after the water passage. The radargrams intersect a network of cracks and englacial channels, meaning we only know its structure at a single cross-section, but the multitude of reflections show more than only one pathway for water. Therefore, we anticipate that elastic contributions are important and cannot be neglected.

Radargrams in 2016 were 11 (and 47) months after lake drainage. In 2018 no massive drainage occurred prior to data acquisition. The profile in 2021 was measured 12 and 24 months after massive lake drainage. Thus, we do not measure right after a gully or cracks are formed. The prominent features we see in the radargrams are all within 850 m above ground. In spin-up simulations of the subglacial hydrological system without seasonality, the head above bedrock is 845-855 m (pers. comm. T. Kleiner, same spin-up as shown in Christmann et al. (2021) ). This supports the fact that the features we see in the radargrams are still active and water-filled. Our radar is not sensitive in the uppermost part of the ice sheet, thus we cannot measure any geometry right below the lake ground.

The modelling results are consistent with our observations of the gully geometry at the surface. We find no change in the area between 2020 and 2022. With increasing depth below the lake ground, the impact of gravitation increases, and the potential to creep-shut the channel rises. End of summer 2020, the head at the gully is still above the lake ground (see Fig. 11), and
in 2023, the head remains close to the lake ground, too. This shows an increase in head due to the drainage compared to the 'background' hydrological system without any perturbation due to lake drainage.

It is striking that several internal features situated in the lower half of the glacier, even very close to the bottom, move without any change in shape or appearance. This indicates a plug-flow regime and no healing during our observational period. Assuming that the farthest downstream feature is indeed from 2012, we infer that even after nine years, no healing took place.
On the other hand, we do find that the most prominent features from 2016 changed their shape by 2018 and 2021.

Gully and fracture locations are concentrated around two locations. One potential reason for this is the influence of bed topography on the local stress field. We find evidence for the reactivation of former drainage pathways. This would fit well with the finding that the gully and fractures are not forming at the same spot every drainage year, but pre-existing failure zones are shifting the actual location slightly.

The principal stresses mainly control the fracture orientation at the surface, and fractures appear as mode I crack, with some deviations due to shear-dominated crack tips. The repeated formation of fractures that lead to a triangular shape for the main drainage channel is not easy to understand from the observations and will need some fracture mechanical modelling studies. GNSS data from 2019 also reveals that the drainage started after the seasonal acceleration had started. This could act as a drainage event trigger if the stresses are already close to a critical threshold, as an increase in strain rate acts as an additional
displacement boundary condition, as was also suggested by Alley et al. (2005).

Striking is that vertical displacement between the crack faces remains over longer time (Fig. 4). To our understanding, drainage events in previous studies settled back to the pre-drainage surface without such permanent shift. The high-resolution



imagery is all taken a couple of weeks after the drainage event (and thus in the phase where the blister is relaxing) and still, rivers flow into the cracks, supplying the basal system with further water volume. This implies that the hydraulic head is below the glacier surface in some years, whereas we find for two years refilling of two gullies which implies that the head is at/above the glacier surface. Chudley et al. (2019) have found large displacement downstream the lake after drainage, while in our case, the largest vertical displacement is inside the lake area (based on subsidence weeks after the drainage).

The farthest downstream reflection in the radargram is a remnant from the 2012 drainage event. The first drainage event in 2005/06 was located further south than the profile we surveyed here. This underpins our finding from the satellite remote sensing data that no drainage occurred before 2005/06.

In just ten years, this system changed from no seasonality to seasonality, and extreme events in the form of massive perturbations of meltwater input with volumes in the magnitude of water reservoirs occurred on a daily scale.

The main question as of now is if the subglacial system is due to the frequent drainage events in transition into a new state, or if it is (still) getting back to a normal winter state despite such extreme water inputs. As further downstream from this lake, massive changes of the system originate still on the grounded ice sheet with forming channels at the underside of the ice (Zeising et al., 2024), the imminent question is, if both developments are linked - and how this system will evolve in future.

## 5 Conclusions

This study is concerned with a massive supraglacial lake at 79NG, which is existing only since the mid 1990's. We find that the formation of this channel is linked to increase in surface temperature in the catchment of the lake. The lake volume exceeded $10^8\,\mathrm{m}^3$ in 2015, but thereafter reactivation of drainage pathways led to smaller lake volumes and drainage happened more frequently. The drainage pathways are cracks and triangular-shaped gullies. Gullies become refilled up to the surface in some years highlighting the head above bedrock to rise to exceptionally high values and remain at this level until freezing occurs. Displacement of crack faces in vertical direction remnant even more than 15 yrs after drainage. We find englacial features that persist over years, but alter in shape and are advected downstream. Englacial features exist only directly underneath the lake or are advected downstream. We find a blister beneath the lake and its neighborhood of non-radial shape, that relaxes over two months.

*Data availability.* We will provide the following primary data: GNSS data (Fig. A1), laser scanner data (Fig. 7, UWB radargrams Fig. 12,13). In addition, we provide derived data such as the crack paths (Fig. 9, 9 ), the outline of maximum filling (Fig. 3a) the location of the gullies (Fig. 3b). All this data will be made available via the data portal PANGEA, and the doi's will be available before publication.





485 **Appendix A: GNSS**

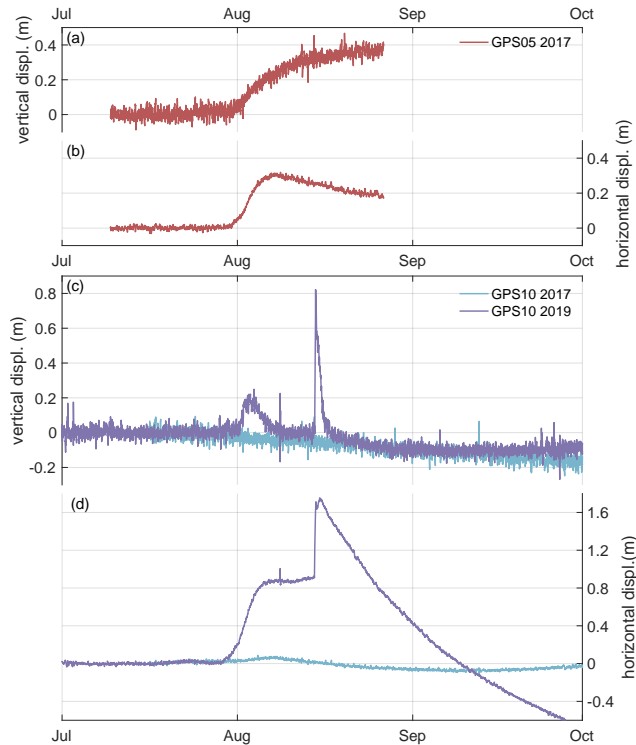

**Figure A1.** Horizontal and vertical motion in the vicinity of the lake. Panels a and b present data from a GNSS downstream of the lake in the summer of 2017, while panels c and d display data in about 5 km distance to the lake in the summer of 2017 and 2019. The location of the GNSS stations is shown in Fig. 1.





## Appendix B: Blister relaxation

**Figure B1.** Double-differential interferograms using Sentinel-1 data. The four panels cover a period from 3-4 to 7-8 weeks after the main drainage. The dashed white line represents the maximum filling on 2019-07-26, while the bold white line represents the cracks on 2019-08-23.



*Author contributions.* A.H. conceptualised and led the study. A.H., V.H., Ma.Bo., N.N. contributed to the methods; data contribution by R.S. and G.E.. Modelling was conducted by Ma.Bo. with supervision by J.S. and R.M.. O.Z analysed CARRA data. S.A.K retrieved the GNSS station data. A.H., H.S. and R.M. conducted the fracture mechanical analysis; M.R. contributed the inverse modelling. A.H. wrote major parts of the manuscript. All authors have discussed the results and reviewed and read the manuscript.

*Competing interests.* The contact author has declared that neither they nor their co-authors have any competing interests.

*Acknowledgements.* The airborne data was acquired as part of AWI's 79NG-EC campaign in 2021, and the RESURV79 campaigns in 2018 and 2016. The airborne campaign 79NG-EC is AWI's contribution to the GROCE2 project funded by the German Federal Ministry of Research and Education under Grant No. 03F0855A. TerraSAR-X data were made available through the German Aerospace Center proposal HYD2059. We would like to thank the crew of Polar 5, Dean Emberly, Marc-Andre Verner, Luke Cirtwill, and Ryan Schrader. the aircraft coordinators Daniel Steinhage and Martin Gehrmann, and the team of Villum Research Station and Station Nord for their support. WorldView data was made available by ESA's Third Party Mission Program. H.S. acknowledges funding by the DFG within the Collaborative Research Center 1313 (Project Number 327154368–SFB 1313) and under Germany's Excellence Strategy—EXC 2075—390740016. S.A.K acknowledges support from the Carlsberg Foundation – Semper Ardens Advance program (grant no. CF22-0628)



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
