# Peer review of "Supraglacial lake drainage through gullies and fractures"

_EGUsphere, 2024_

## Author Comment (AC1)

Dear Reviewer #1,

we want to thank you for your very helpful comments on our manuscript, that we have addressed in the detailed answer below. We are looking forward to your comments on the revised version of the manuscript.

Best wishes,
Angelika and all co-authors

**RC1**: 'Comment on egusphere-2024-1151', Anonymous Referee #1, 04 Sep 2024

The manuscript, 'Supraglacial lake drainage through gullies and fractures' by Humbert et al. seeks to characterize the ice surface and englacial behavior associated with repeated supraglacial lake drainage on Nioghalvfjerdsbræ Glacier using a wide range of in situ, satellite and airborne observations.

Overall, I found the individual observations interesting – particularly the radargrams. However, I believe that there are several fundamental issues with the manuscript. My major concerns are detailed below followed by line comments.

1.  The manuscript is missing a story. Many datasets are presented, but by the end, I am still unclear about the manuscript's main conclusions. Those indicated in the final section of the paper aren't well supported by the analysis presented.

    This point was also raised by the second reviewer and we take it seriously. We have got stuck in so many details of the data, that we lost in the text the focus of the story. We will follow your suggestion as well as the one from RC2 and reduce the introduction to englacial features in Greenland, elaborate on the findings of other studies and than introduce our approach.

    We do think that our conclusions are very well supported by our analysis, but we admit, that we failed in presenting this in a proper manner. This will be improved in the revised manuscript.

2.  Why not term these moulins? Particularly after they start receiving persistent meltwater and reactivate over time? I understand the argument that these structures aren't cylindrical (at least at the surface) but drainage to the bed along cracks is certainly a moulin. The imagery presented here are very similar (at least in my mind) to figures in Hoffman et al. (2018), Chudley et al. (2019), and Doyle et al (2013). Yes, the englacial geometry is complex, but that is not unexpected (e.g. Covington et al. 2020). Plus, I'll note that even Das et al. (2008) refer to established surface-to-bed flow as 'moulin flow'. These features are moulins and calling them something else complicates the story, particularly when modeling the deformation of a circle at the surface.

    We understand that the term gully is not very welcome. The terminology of hydrological features of ice sheets has evolved over time, not only with respect to moulins, but also other features. We suggested, that it is time to move away from the term moulin for any kind of drainage pathways and if one wants to keep a text consistent, moulin fracture is leading to substantial longer text. And indeed the englacial geometry is expected to be complex, but is it governed by melt features, like scallop shaped surfaces of moulins walls? We do not

expect this to be the case. If water flows through moulins that are formed by melt features 'moulin flow' makes a lot of sense. But does the same term makes sense, if fractures of km's length open, water drains through in short time (e.g. Chudley et al. (2019), and almost closes again? The englacial features in both cases are of different nature. So, why not using a distinct terminology for both to make it more clear?

We want to give some background on why we have chosen this distinct terminology: gullies in streets are pathways for rain water into the underground sewage system. Often their cover acts as a mesh and after torrential rain, dirt, branches etc are filtered, staying on the surface while the water drains into the sewage system. Similar characteristics is shown in Fig. 5 c with ice blocks not drainaing through the conduit. When the sewage system is itself flooded, the water level raises and water is rising onto the street. Similar to what we found and visualied in Fig. 11. Lastly, the manhole cover is displaced when water raises from the sewage system and is lying close to the manhole, similar to the blocks we found. These are a number of similarities.

What is also clear to us: if both reviewers are rejecting the terminology so clearly, there is no chance that the community will use it. Which is not useful at all. Therefore, we surrender and will rephrase the text accordinly to moulins.

3. The analysis of the englacial features is limited to the visual inspection of a series of radargrams. It would be preferable to more carefully analyze these unique data and explore how the results link to surface behavior of both the supraglacial lake and crevasses/cracks.

We do see your point and we will try our best, to give more details in this regard. But we want to stress, that the radargrams are acquired not straight after a drainage event. None of the airborne campaigns 2016, 2018, 2021, recorded the englacial features as they were formed, but at a later stage of existence. Our airborne flightlines were also not so densly spaced, that we could aim for a 3D reconstruction of 'the real' structure. This is something we anticipate for the future.

For the revised version, we will enhance the discussion of the radargrams and we will also include a discussion of the limitations this surveying is objected to.

4. Not all the methods are described in the Methods section. The manuscript references GNSS derived ice velocities, inverse modeling, methods for displacement along a crack and subglacial modeling. In these cases, the methods are not clearly described. In other cases, where the method is described, there are missing references to the tools or software used.

We will extend the method section in the revised version and include for each method an own section. We will also make sure tht references to tools and software is included.

5. There is an overall lack of referencing, particularly in the discussion. There has been a reasonable amount of recent work looking at the mechanisms, geometry and impact of moulins on the subglacial hydrologic system. I have included some of these references below, but there are a number of others.

Many thanks for supporting us with the literature! About one third of the suggested papers were indeed cited in our manuscript and we are very happy to include more in the revised version.

6. The manuscript would benefit from a through proof reading, including grammar, punctuation and abbreviations.

Many thanks for pointing this out. We will work on grammar, punctuation and consistent use of abbreviations in the revised version.

L3 The first sentence is missing something.

Indeed, this is correct. We have completed the sentence now.

L30 The literature indicates that additional meltwater can both act to accelerate AND slow basal sliding depending on the structure and evolution of the subglacial hydrologic system. It would be worth clarifying that here.

This is a very good suggestion! We will cite the following paper in the revised version and add a sentence to clarify this.

Moon, T., I. Joughin, B. Smith, M. R. van den Broeke, W. J. van de Berg, B. Noël, and M. Usher (2014), Distinct patterns of seasonal Greenland glacier velocity, Geophys. Res. Lett., 41, 7209–7216, doi:10.1002/2014GL061836.

L38 How does lake overflow differ from feeding downstream lakes and streams?

We wrote 'Supraglacial lakes may either drain straight into moulins, or partially by overflow, or by feeding lower elevation lakes via rivers and streams.' not 'feeding lakes and streams' . Maybe there is a misunderstanding? We will reformulate this sentence and include here also two of the suggested papers below.

L40 Consider revising this section. I think the general mechanism of lake drainage is reasonably understood – in compressional lake basins, there needs to be a precursor tensile event (e.g. Hoffman et al., 2018; Stevens et al., 2015; Christofferson et al., 2018). What causes the tensile event is still up in the air and can vary from place to place and because these events are difficult to observe (Poinar and Andrews, 2019) there are outstanding questions.

We disagree with this comment. As long as the cause of the tensile event is not clear, the drainage mechanism is not understood. And also the cause of the triangular shaped features in our case or more arch-shaped features in Chudley et al. (2019) is not clear at all.

L45 "In Greenland, there are…"

Done.

L45 Gulley et al. (2009) provides a nice review of the mechanisms described in this paragraph and it would be beneficial to include some of the information included there.

This is a very good suggestion and we will follow this in the revised version.

L55 The primary difference isn't just scale. The current understanding is that most of the englacial structure in Greenland is formed via hydrofracture. Other mechanisms like cut and closure struggle due to ice temperatures and overburden pressures.

Not so in crevasse fields.

L85 Include a table of the satellite, image names, dates collected and resolution. Table 1 is close but doesn't have all the information to be reproducible.

To include the resolution in the tabel is a phantastic idea! We will do that for the revised version. As we literally checked each existing Landsat and Sentinel-2 image, there is no more information drawn from listing the dates of all of those. The other review suggested to turn this table into a figure. We will draft a figure and check, if that is better digestable than a table with even increasing information content.

L165 So, reading the CARRA documentation, it looks like skin temperature is "Average air temperature at the surface of each grid column." Which is different than the temperature of the uppermost surface layer – which shouldn't respond instantaneously to surface fluxes. Some clarity here would be beneficial.

Following the CARRA documentation (Parameter Database), the skin temperature is "the temperature of the surface of the Earth. The skin temperature is the theoretical temperature that is required to satisfy the surface energy balance. It represents the temperature of the uppermost surface layer, which has no heat capacity and so can respond instantaneously to changes in surface fluxes."

L170 It would be beneficial to have a cross-sectional diagram as well as Figure 2 or clarity that the model is only for surface deformation. Also, this modeling framework is in direct contradiction to the argument in the introduction about not being circular, thus not being a moulin.

In L70 the modelling part is introduced as a case study for a drainage channel at the surface of a glacier. In L170 the model is described as a two-dimensional top view on the englacial channel, with a visualization of the model in Figure 2. However, the phrase "evolution of the englacial channel" may be to abstract and we will work on a more specific formulation and provide more details on the assumptions of the numerical model. In the revised version, we will clarify our aim to simulate the radial displacement of the channel on the top surface.

L213 Table 2 would be better as a figure!

This is a very good idea and we will consier including this into Fig. 3!

Further, how is the 'begin filling' and 'filling complete' dates determined? It seems that these would be difficult to determine and the 'filling complete' would just be the date that drainage started.

Filling complete has been determined from stagnant area of the lake.

L255 Inferring that the drainage paths were shut should be in the discussion, not the results.

This is a very good suggestion, which we will follow for the revised version!

L241 This is the first mention of ice-based GNSS position measurements. These measurements and the processing to velocities should be described before this section. 10% variation seems small when looking for the addition of meltwater to the bed. Also, the inference that there is meltwater at the bed should go in the discussion.

We added in the method section a subsection were we introduce the ground based GNSS data.

"To measure horizontal and vertical displacement of the glacier surface through time two dual-frequency NovAtel GNSS receivers were deployed in the 2017 field season. The

receivers collected data for roughly two years with some data gaps during winter time. The flow trajectory was computed using the precise point positioning (PPP) post-processing option, which included precise clocks and ephemerides of the commercial GNSS software package Waypoint 8.90. The accuracy of the post-processed trajectory is better than 0.02\,m. Based on the trajectories we were able to estimate temporal variations in the ice flow velocity and vertical displacements after lake drainage events."

We agree hat the inference should be moved to the discussion and will do so in the revised version.

L255 This figure reference is out of order.

Yes, indeed the reference to the figure is incorrect. Many thanks for pointing this out!

L268 (& L289). How are ice block widths measured? Using 10-m Sentinel-2 imagery would not permit widths less than 10 m.

This has been measured in the WorldView imagery.

L274 If the 2020 'gulley' was still active, drainage did take place. Do you mean to indicate that no 'rapid' drainage took place?

Yes, indeed this is what we wanted to express. We are rephrasing this in the revised version to make that clear.

L298 These sentences are interpretation best left for the discussion.

We agree and shift this into the discussion section in the revised version.

L317 What is meant by 'shade'?  Shading?

Yes, the dark shadowed area of an elevated obstacle. We will rephase this to shading.

L336 What inverse modeling? Such methods should be described in Materials and Methods. What velocity fields are used? This choice would drastically affect the derived stress fields.

This is a very good suggestion. We wll include this in the revised version as an own method section, give information about the velocity fields used. As a comment, the velocity field has not changed substantially in this area since 2008/09. So it will have at most a minor effect on the principle stresses used here. In other regions around Greenland, also in the close neighbourhood this will be very different.

L345 without more details about the inverse methods and associated choices, this statement is speculative.

As stated just above, we will definitely include more information in the revised version of the manuscript.

L347 Is there other evidence that Figure 10 actually shows uplift? In this case, I would expect one edge to be sharper. What is the sun angle and orientation?

Which edge should be sharper? One needs to keep in mind, that the surface of the ice before the fracture appeared inhibited some roughness, too. Therefore, we do not understand where the sharp edge should come from.

In the revised version the information on sun angle and orientation will be given.

L419 The modeling results need further description and justification. Is this meant to be a vertical cylinder in the ice? horizontal? Why only run the model for 20 days when the time between lake drainages can be several years? The figure seems to show results at or near the surface because there would be substantial creep closure at depth, but it is unclear to me if the englacial conduit is modeled as water filled or air-filled.

This corresponds well with your comment on L170. We are going to work on a clarification for the geometrical setting, as we try to simulate the surface deformation of a vertical cylinder. We simulated the deformation of the channel within 20 days since the modelling framework holds only for small deformation. However, the tendency to open is still predictable with small deformations during the immediate drainage event. Therefore, we chose to model the channel with 10 days water-filled and 10 days air-filled. Clarifying sentences will be added in the revised manuscript.

L430-435. This paragraph is not well justified and the reference an undefined subglacial model needs further description. I think it is conceivable that the water table within the conduits is identifiable in Figure 12a-b, but additional, careful justification is needed. Subglacial models are notoriously poor at capturing observed subglacial pressures, particularly if they do not include point supraglacial inputs and there is no modeling to support that an englacial conduit could remain open in the absence of supraglacial water inputs. I will note that Figure 12c could indicate a water filled moulins, which can be quite complex (e.g. Covington et al., 2020).

We did not intend to use this information to draw conclusions on the water pressure during the drainage events and we fully agree with the reviewer that our model does not capture the water pressure correctly during the drainage or shortly after. We intendend to use this as a plausibility check. And for this purpose, we find it interesting and useful information. But in order not to confuse readers, we will leave that out in the revised version.

L437-441 This paragraph is unclear. Consider rewriting.

We agree and will rephrase the entire paragraph in the revised version.

L442-444 The reflections in Figure 12 would be due to the difference between ice and air or ice and water – it's unclear how these reflectors can provide information about whether the moulins (or englacial conduits) have experienced any creep closure or thermally driven opening.

We do not say anything about creep closure or the reflection coefficient air/water versus ice/water. We simple compare the position of identified features over time. And this clearly shows displacement is not depending on depth.

L446 This behavior is consistent with moulin behavior – moulin reoccupation is common and dictated by surface gradients and moulin advection. However, what evidence justifies the statement of reoccupation? Figure 12? Why?

The text in L446 states that new gully features are created at the same locations (denoted A and B). Once formed they are advected, but this sentence just explains that they are created repeatedly at the same locations. Figure 3b justifies this.

L460 Or it could be that the moulins/gullies closed at depth between drainages and needed to fill in order to hydrofracture and reactivate.

Fig 11 shows the time scale at which this refilling is happening. We find it extremely unlikely that the closure happens on such a short time scale and that sufficient water supply is available (no water around the gully location in Fig. 11) to fill a crevasse in this form.

L466 The skin temperatures display a clear seasonality from 1991 (Figure 15). What exactly is meant here?

We meant seasonality with respect to formation of lakes, drainage and blister formation and the respective influence on the subglacial system. We will clarify this in the revised version.

L473 The statement that the englacial channel is due to increased surface temperatures is not justified here. Perhaps there is more frequent lake drainage due to higher meltwater production.

Correct, there is a sentence missing in the discussion that connects the first drainage with atmospheric forcing.

Figure 3. In panel b, what do the different orientation of the triangles mean? This isn't described.

There is a legend included in the Figure which clearly defines each symbol

Figure 4. What is the color scale in panel a?

Panel a shows a Pauli decomposition of a quad-pol image, which is a standard in satellite polarimetry. This question is similar to what is the color scale of an optical satellite image with RGB channels.

Figure 5. Having the panels and regions be the same letters is confusing.

We cannot really see why this should be confusing. The regions are denoted with A, B and the panels with (a), (b) etc.

Figure 7. No panel letters. The spatial resolution and orientation of panel c? is different. Why?

We will include panel letters in the revised verson. The orientaton is the same. We have zoomed into the area, to allow the reader to see the elevation in the triangluar-shaped area better. But we are happy to keep all figures the same scale.

Figure 12. What are the F's, T's and W's? They aren't referenced in the caption or text.

In the figure caption we added some more information:

"We identified englacial features (labelled as F1 to F5), which reoccur in all three panels (F3) or newly develop (F4, F5). A persistent wavelike feature is labelled as W1 and a triplet of features close to the ice bed are labelled as T1."

Figures 12 and 13. Can the flight lines be added to a map?

The flightlines are shown in Fig 1 and the caption of Fig 12 and 13 describe that they are shown in Fig. 1.

References

Andrews, L. C., Poinar, K. and Trunz, C.: Controls on Greenland moulin geometry and evolution from the Moulin Shape model, The Cryosphere, 16(6), 2421–2448, doi:10.5194/tc-16-2421-2022, 2022.

Christoffersen, P., Bougamont, M., Hubbard, A., Doyle, S. H., Grigsby, S. and Pettersson, R.: Cascading lake drainage on the Greenland Ice Sheet triggered by tensile shock and fracture, Nature Communications, 9(1), 1064, doi:10.1038/s41467-018-03420-8, 2018.

Chudley, T. R., Christoffersen, P., Doyle, S. H., Bougamont, M., Schoonman, C. M., Hubbard, B. and James, M. R.: Supraglacial lake drainage at a fast-flowing Greenlandic outlet glacier, PNAS, 116(51), 25468–25477, doi:10.1073/pnas.1913685116, 2019.

This paper has been cited and discussed

Covington, M. D., Gulley, J. D., Trunz, C., Mejia, J. and Gadd, W.: Moulin Volumes Regulate Subglacial Water Pressure on the Greenland Ice Sheet, Geophysical Research Letters, 47(20), e2020GL088901, doi:https://doi.org/10.1029/2020GL088901, 2020.

Das, S. B., Joughin, I., Behn, M. D., Howat, I. M., King, M. A., Lizarralde, D. and Bhatia, M. P.: Fracture Propagation to the base of the Greenland Ice Sheet during supraglacial lake drainage, Science, 320(5877), 778–781, doi:10.1126/science.1153360, 2008.

This paper has been cited and discussed

Doyle, S. H., Hubbard, A. L., Dow, C. F., Jones, G. A., Fitzpatrick, A., Gusmeroli, A., Kulessa, B., Lindback, K., Pettersson, R. and Box, J. E.: Ice tectonic deformation during the rapid in situ drainage of a supraglacial lake on the Greenland Ice Sheet, The Cryosphere, 7(1), 129–140, 2013.

This paper has been cited and discussed

Gulley, J. D., Benn, D. I., Screaton, E. and Martin, J.: Mechanisms of englacial conduit formation and their implications for subglacial recharge, Quaternary Science Reviews, 28(19–20), 1984–1999, doi:10.1016/j.quascirev.2009.04.002, 2009.

Hoffman, M. J., Perego, M., Andrews, L. C., Price, S. F., Neumann, T. A., Johnson, J. V., Catania, G. and Lüthi, M. P.: Widespread Moulin Formation During Supraglacial Lake Drainages in Greenland, Geophysical Research Letters, doi:10.1002/2017GL075659, 2018.

Poinar, K. and Andrews, L. C.: Challenges in predicting Greenland supraglacial lake drainages at the regional scale, The Cryosphere, 15(3), 1455–1483, doi:10.5194/tc-15-1455-2021, 2021.

Stevens, L. A., Behn, M. D., McGuire, J. J., Das, S. B., Joughin, I., Herring, T., Shean, D. E. and King, M. A.: Greenland supraglacial lake drainages triggered by hydrologically induced basal slip, Nature, 522(7554), 73–76, doi:10.1038/nature14480, 2015.

This paper has been cited and discussed

**Citation**: https://doi.org/10.5194/egusphere-2024-1151-RC1

---

## Author Comment (AC2)

Dear Reviewer #2,

we want to thank you for your efforts to improve our manuscript with your comments. Below you will find detailed answers. You raised some points that were also raised by RC1 and your might be interested in our answers to RC1 as well.

Many thanks again,

best wishes,

Angelika and all co-authors

**RC2**: 'Comment on egusphere-2024-1151', Anonymous Referee #2, 05 Sep 2024

Humbert et al. present a detailed study into the long-term drainage history at a previously unreported lake at N79 glacier in northeast Greenland. The study integrates a wide-ranging array of highly technical methods with a high degree of competence. The paper is dense and technical, and clearly the work that went towards it is commendable. The two most novel parts of the study, in my opinion, are: (i) the observation of a system than, within the satellite observational record, developed from no lake at all into a lake that displays repeated rapid drainage to the bed; and (ii) the first (as far as I am aware) radar observations of the annual fracture history caused by repeated lake drainage.

My comments on the paper largely align with that of Reviewer #1. I feel that my understanding of the paper was limited by some strange terminology choices and a dense and confusing structure, which has made me feel as though I might be missing the main thrust of the story. I hope that this can be fixed largely within the terminology and structure, rather than implying any deficiencies in the work itself.

**General Comments**

It is a key part of the study, but I am still unclear as to what the authors mean when they refer to 'gullys', beyond the general dictionary description of a water-incised channel. I initially read the paper following my interpretation of the dictionary definition, whereby the 'gullies' are the three drainage channels extending E-NE from the three 'branches' of the lake (e.g. Fig. 1a), and 'drainage through gullies' would refer to the lake overtopping and draining due to the incision of these channels (as described in Tedesco *et al.* 2011, doi:10.1088/1748-9326/8/3/034007). However, as I read the manuscript, it became increasingly clear that 'gullies' appeared to refer to point features that might be analogous to moulins. Most obviously this includes: Figure 3b (as triangles); L236 ("a hole… is the first feature we identify as a gully"); L254 ("a gully [can be] inferred due to ponding … which has a form consistent with a drainage pathway"); and Figure 7 (the 'gully' here seems explicitly to be the point depression). Gullies are also frequently referred to as 'triangular', although it is not clear in what context (plane? profile?).

We think, Figure 7 does indeed show a triangular shaped feature and all panels show either satellite imagery or airborne data and hence a top view of the horizontal plane. However, if this did not become clear, our text lacks a proper introduction of that, which we will include in the revised version.

To me, if these point features are what the authors are referring to, I cannot see why they are not referred to as moulins. The authors take time to explicitly seem to reject this in paragraph beginning L75 as 'are neither formed by melting nor round in shape'. I have never encountered the necessity that a moulin must be formed by melting, nor strictly round (beyond the fact they can be generally modelled as point rather than linear inputs to the en/subglacial system). Post-drainage surface-to-bed (or, at least, surface-to-englacial-environment) connections are commonly called a moulin in the drainage literature. Looking at the drainage pathways visualised in the study (Fig 5a; 7; 8c, 8d), I cannot see how they are different from those referred to as 'moulins' in previous drainage studies, which also show rivers terminating into thermomechanically-maintained holes along the relict fracture (Das *et al.* 2008, Fig. 1 inset, doi:10.1126/science.1153360; Doyle *et al.* 2012, Fig. 4a, doi:10.5194/tc-7-129-2013; Chudley *et al.* 2019, Fig. 5e, doi:10.1073/pnas.1913685116).

We have included a more general discussion on gully versus moulin terminology in our answer to RC1, so we want to focus here on the last parts of this comment:

In the paper of Das et al. (2008) it is not easy to see clearly the fracture geometry, but what is clear from that figure is that the shape is not triangular and by far lager in size than what we denoted gully. But the Fig1 inset there and the Doyle-Fig4 and Chudley-Fig5 are significantly different. The Das-feature is neither triangular nor arch-shaped as the Doyle and Chudley features. We found in numerous instances at other lakes fetures similar to Doyle's and Chudley's.

If I am right that the gullies are the point features, then - to me - these drainage events bear a close resemblance to previous events: (i) background fractures exist oriented according to the principal stress (L337); (ii) which, aided by transient flow acceleration (L454-455), the water can then exploit (or reactivate) via hydrofracture (as in Christoffersen *et al.* 2018, doi:10.1038/s41467-018-03420-8). The authors say that hydrofracture isn't occurring at L78-79, although I can't see any evidence to reject Occam's razor, especially as no other mechanism is proposed for the 'propagation' referred to at L343-346); (iii) this fracture opening results in drainage to the bed (or, at least, englacial environment); (iv) full-depth fracture will close elastically along most of the length (L426-427); (v) apart from where rivers intersect the fractures (L334) and surface-to-bed connections (moulins) can be maintained through thermo-mechanical erosion of the vertical drainage pathway. If this is the case, then 'gully refilling' may be similar the relict moulin refilling previously noted by Chudley *et al.* (2019) in the literature (cf. Fig 11 of the manuscript with, e.g. Fig 5c insets in Chudley *et al.*), although they attribute this overtopping to the 'top-down' filling of a closed moulin rather than 'bottom-up' water rising from the bed in this paper.

When we stated that the fractures that lead to drainage are not formed by hydrofracture, we intended to explain, that the overall stress situation of this area is leading to initiation of new fractures. The background fracture are shallow and narrow. But the stresses governed by bedrock topography changes are in our perspective the ones who are governing the location of fracture formation.

I do not think that similarity to previous studies is a bad thing: instead, the choice to choose different (and poorly explained) terminology for relatively ambiguous reasons is confusing when trying to place this study into the context of other work (at least, it is for me!). If I have been misled by my interpretation, then perhaps the authors might need to take more time to more carefully explain the new terminology, and how and why this lake drains differently from previous studies.

We have elaborated on our view of this choice of terminology at below and in the answer to the other reviewer in detail and kindly like to ask to consider those answers, too.

**Specific Comments**

[Abstract] Include '79˚N Glacier' as alternative name within abstract if there is space?

Yes, this shall fit and is certainly useful for the reader.

[Introduction] I found it surprising how dense some of the introductory material is, covering quite technical aspects of flow dynamics, englacial hydrology, and fracture mechanics that are not touched upon again in the rest of the paper. Some work could be done to remove unnecessary content here. Perhaps this relates to Reviewer 1's comments about a lack of clear story - a more cohesive set of key findings would in turn help to identify the necessary material for the introduction.

Many thanks for this feedback. Yes, we agree, this could definitely be shortened and more streamlined to clarify the story. We will work on this intensively for the revised version.

[L24-27] Is this level of detail necessary for the introduction? Could just say that AF is well-established to have occurred in this region (citing Khan, Humbert, Zeising).

We certainly can reduce the four lines to three lines if that is helpful.

[L27] 'In this study' - perhaps a misrepresentation of what this study is doing? It is perhaps not surprising to find how rising surface temperatures could lead to a meltwater lake.

We agree that this sentence was missleading. We rephrased it to: "In this study, we will show that the formation of a massive meltwater lake coincides with changes in surface temperatures."

[L29-35] Given the well-established lake literature (and how none of this really comes up in the rest of the paper?), this is a lot of words to say that lakes occur where surface melt collects in topographic depressions.

Line 29-35 discuss how runoff occurs and how local melt can straight reach the bed. In these six lines, lakes are not the topic.

[L75-78] I am quite confused as to why this text is located here, in between to other completely standard final-paragraph-of-the-introduction prose.

We intended to reiterate the terminology, but we will revise this text.

[L85-92] I do not think this section needs to be three separate sentence-long paragraphs. This applies elsewhere as well - the abrupt appearance of short interspersing pargraphs gives the impression the sentences/paragraphs have been cut-and-pasted together without much regards for coherent structure or flow.

We apologise and will edit the text accordingly. It was not meant as individual paragraphs.

[L94] Perhaps 'single-polarization (hereafter single-pol)' at the first instance.

Many thanks for this suggestions! We will change the text accordingly.

[L103] Was there significant difference in the relative basins (barring vertical difference due to melt, presumably) between the three DEMs? This would be of interest as, if the three

DEMs gave broadly similar results, it would mean that long-term lake volume studies could be produced from confidence using only one good DEM of the empty lake basin (from e.g. the ArcticDEM mosaic product)

Thank you for raising this point as indeed this would be nice for global studies on supraglacial lake volumes – but unfortunaty the shape of the basin changed quite a bit between the three time stamps. To visualize this we show the 850m contour line derived from the three different DEMs. As evident from this image using a single DEM would also have a large impact on the derived lake volumes.

[Figure]

[Section 2] Could this be further separated into 2.1 Satellite Methods, 2.2 Airborne Methods, and 2.3 Modelling for clarity, with current subsections as subsubsections?

Yes, this is a very good idea and we will include this in the revised version.

[Table 1] could be moved to a/the supplement for brevity

We recognised that R1 even wants this table to be extended with even more information and we found within the team of co-authors that it is useful to flip back- and forth between the figures/text and this table. Our suggestion is to leave it with slightly more information (resolution) in the main part of the manuscript.

[L211-217] could be methods?

This is indeed a great suggestion that we will incorporate in the revised version.

[L223-4] Once again, is this really its own paragraph? I note at this point in the results there start to be single-spaced line breaks and double-spaced line breaks. This is again confusing - what are paragraphs and what aren't?

We intended to separate the years as paragraphs, but in that particular year, little information could be gathered, so it is extraordinary short. But you are definitely right and we will rearrange this in the revised version.

[P13] Could a timeseries plot be a useful way of visualising this narrative? Noting e.g. maximum extents per year, with drainages marked.

This was also raised by RC1 and we fully agree that this makes a lot of sense. We are working on a draft version of such a graphic, also in order to replace the table.

[L231] Here and elsewhere, satellite imagery used to build the narrative is not shown. Perhaps as supplementary material, before/after satellite images of the drainages could be visualised?

This is definitely an option. We felt like the manuscript is already overloaded with imagery, but putting it into supplementary material is a very good idea. We will prepare this for the revised version.

[L253-257]. This is the first point in the text that locations A and B are referenced, and after finishing the manuscript I am still unclear as to exactly why the specific locations are important or what is happening in them that is special. Is it where the 'gullies' (moulins?) are located? Are they the same ones being reactivated and advecting along? This needs to be much more clearly explained. Also, it is confusing to track them between figures - can they be marked on all of them?

This is 46 lines into the results section and is in the section where the time period is discussed in which it can first been narrowed down to such a location. It has also been discussed and shown in Fig3b what why we mark this. We will go through the text again and check carefully where we can add more text to make it easier for the reader to follow.

[L258] 'Gully A' referenced here and, as far as I can tell, never again.

Indeed – this is a relict and we have changed that for the revised version.

[L293]. and elsewhere - present tense rather than past tense.

We are going to check this throughout the revised version. Many thanks for pointing this out.

[Section beginning L360] I agree with Reviewer 1 that the aerial data is remarkably lightly analysed considering - as I identify in my opening paragraph - it is truly spectacular and unique data with a wealth of opportunity within.

We fully understand the wish of the reviewers to apply more analysis on the radar data. We are exploring ways to enhance our analysis and present this in a useful way for the reader in the revised version. We will also include a discussion of the limitations of the radar surveying and its interpretation.

[Paragraph beginning L430] Other studies that discuss the elastic opening/closure of full-depth crack closure/opening suggest it must be dependent on continued flow of water to remain open, and closes rapidly after the main drainage event is complete (e.g. Doyle et al, Chudley et al, Stevens et al). However, here it is implied that, with just the hydraulic head alone, (i) full-depth cracks can remain open; and (ii) water does not freeze (cf. e.g. Hubbard et al 2021). My instinct here is that this is quite unrealistic without further evidence?

The simulation results in Figure 16 show that after removing the water pressure from the boundaries of the cylinder, the radial displacement of the boundary still increases, with a positive mean value. This can be attributed to the different boundary conditions at the outer boundaries of the simulation domain. On the left side, we applied the upstream velocity of 3.8 m/d and on the right side the downstream velocity of 3.9 m/d. Both velocities are in accordance with remote sensing observations, as stated in L203. Because of the velocity difference, the simulation domain not only faces a rigid body motion, but an elongation, which appears to be a driving force for the opening of the channel in this model. We will incorporate this in the revised version.

[L433 - 441], and potentially elsewhere: does 'head' refer to the hydraulic head?

Yes, indeed head refers to hydraulic head. We will use consistently the term hydraulic head in the revised version.

[L456-457]- Post-drainage vertical displacement along the crack face was also reported by Doyle *et al.* (2013)

Many thanks for raising this. We will incorporate this into our revised version!

[L455-465] I agree that it is likely the 2005/06 event is likely the earliest event (within the satellite observational record), but surely the radargram data merely confirms that no drainage occurred from the date of the farthest-downstream-observation of the radargram through to 2012?

We assume that the reviewer refers to the last two sentences in this paragraph. Indeed the radargram is not the strongest argument. We flew in our airborne surveys along a centrally located flow line and the farthest downstream feature fits perfectly well with flow speed and time between the airborne survey and the drainage even. However, we might have just missed a feature that is slightly offset to this line. We will remove the last two sentences.

[Figure 3] maximum lake extent appears to be getting smaller through time? Is this significant?

Indeed this is significant. We shall incoporate this in the results discussion section in the revised version.

[Figure 3b] do the triangles represent the current (advected) locations of the gulleys, or the contemporary drainage locations? Triangle directions aren't explain within caption and require reference to main text.

The triangles represent the location of gullies at the time given in the legend.

[Fig 8a]  The shades of the principal (stress? strain?) crosses are very hard to differentiate easily. Perhaps colour-blind-friendly colour contrast would be better here?

Many thanks for raising this point. As one co-author is colour-blind, we were able to test the suitability of the color choice directly and the result is that the diffence in shades of the crosses is visible.

[Figure 14]  explicitly label the panels "3-4/7-8 weeks after drainage" as well as the interferogram dates for clarity?

This is a very good suggestion that we will incorporate in the revised version.

**Citation**: https://doi.org/10.5194/egusphere-2024-1151-RC2

---

## Editor Decision (ED1)

Dear Prof Humbert and co-authors,

Thank you for your thorough response to the reviewers' suggestions. I find the paper much improved in structure, particularly in the methods and results. However, I agree with the reviewers that the exciting story is rather lost in some of the detail, and despite your rewrite I still find this to be the case. I would like to request another revision, particularly addressing the discussion. I provide some suggestions below, both in detail (including some language and structure edits), and in overview of the main message.

Please provide a revised document, and I will determine whether the reviewers should be engaged again after this revision. We all agree that this is a very interesting study, but that the storytelling needs to be clear to ensure that the paper has the impact that it deserves.

Thank you for your submission to The Cryosphere.

**Suggested amendments**

Note that line numbers refer to the authors' tracked changes document.

L21:'is also playing' – replace with 'plays'

L29: 'the dynamics of ice sheets and outlet glaciers is gravity-driven lubricated flow' – this sentence is very awkward.

Suggest beginning this para at L30 with 'Meltwater that is locally formed in crevassed areas *of the ablation zone* is transported through crevasses to the glacier based and can lead to a seasonal acceleration in ice flow *via basal lubrication*.' Inserting ' of the ablation zone' and 'via basal lubrication' within this sentence to account for the first two sentences.

L32: rephrase: 'However, seasonal glacier acceleration is not linear, but varies according to ice dynamic processes and the behaviour of the subglacial environment (Moon et al)'

L34: combine sentences and simplify: 'Further upstream and at higher elevation locations, the surface meltwater either percolates into the firn matrix, or when melt onset is too rapid to accommodate complete percolation, surface runoff. The runoff becomes an organized system of streams and rivers that lead water to topographic depressions where supraglacial lakes may form. These depressions *usually* correspond to basal topographic lows, and the lakes may range in size from a few square metres to tens of square kilometres. We focus this study on....'

L43-45: remove 'moulins are features that are conduits allowing water passage to the ice sheet base' – if readers have got this far they know what a moulin is.

Add '' around 'moulin fracture' or italicize to denote your terminology, and correct 'tenth' of metres. Should read 'We use 'moulin fracture' as terminology for conduits that are formed by cracking, with horizontal extent of tens of metres'

L49: incorporate the Neckel ref into the previous sentence rather than adding a redundant short sentence.

L51: unclear whether 'these events' is referring to the deep pre-existing cracks or shallow cracks, or both. Please improve.

L61: 'other studies' but only one is cited. Suggest 'it has also been observed that an englacial conduit can remain open over a longer period of time and potentially influence basal melting (Catania and Neumann)'

L71-78: is this paragraph necessary for your story?

L85: remove 'before' at the end of the sentence, since you already have 'previously'

L86: remove 'we selected this one because' – it is implicit in the sentence without the extra words. Sentence then should read 'The lake on which we focus is exceptional in size, but mostly importantly, we were able to build an extensive database from longer-term (1995 to 2023) observations'

L109: remove 'triangular' here since it is the first appearance, and the shape is not important for the identification stage. Or define 'triangular moulins' here.

L113: some formatting errors here!

L185: sudden switch to passive voice (this study uses, vs. we used). Suggest being consistent.

L256: the first sentence is tricky for a non-modeller to interpret. What are the reasonable principle angles? A full definition is likely not required, but perhaps the sentence order could be swapped so the para begins with 'We leverage the ISSM...', or add a few words at the end of the sentence to improve understanding 'We employ an inverse model similar to..... in order to obtain reasonable principle angles *of.....*'

L291: 'cracks were formed' instead of 'have been formed'

L293: Sentence beginning 'End of August' is missing 'At the' at the start: 'At the end of August, an overflow....'

L294: 'which we do not display here' – will these be in supplementary? There's a few of these references throughout the paragraph. I wonder if its useful to say 'not shown' since it leaves the reader feeling short-changed?! Instead, you could just state your observations.

L316: do you need to note that the feature is triangular here? You use 'triangular' 11 times in this para – I think a few instances can just be 'moulin'

L326: formatting error

In this section, the placement of text and figures occasionally makes the story harder to track, but I trust the editorial support team will come up with some good solutions on formatting of the final article

Fig 7 caption: 'Panel a and b show' instead of 'are showing'

Page 23-7: the reviewers note inconsistent paragraph structure and this remains unaddressed, particularly here. Please revisit the structure of these pages to group the sentences into thematic paragraphs.

Figure 10: not sure how useful this one is

Figure 13: add year of acquisition to the caption, and remove final m from radargram

Figure 14: define units of colour bar

Discussion

The first paragraph of the discussion needs to give an overview of your findings to help us understand a) what you have discovered and b) why this is important. At the moment, it launches straight into detailed explanation of process, which doesn't help the reader appreciate how all your different lines of evidence meet together, and why this is a significant finding for the discipline. Why is this event unusual? It may be that this is the most complete observation set for a cyclical drainage event of this size, and that the feature persists over time.

I do not find that the discussion as rewritten helps address all the reviewers' concerns, therefore I would like to request another rewrite that focusses on:

1. Explaining your story with a link between the different types of observations that you used and the processes which explain them (L536-559 do a really nice job)
2. Giving an overview of how this feature fits into our knowledge of supraglacial drainage events
3. Ensuring the paragraph structure is consistent

L581: which channel?

L589: the conclusion finishes abruptly. I request a final sentence that stresses the importance of the study, or the main finding.

Dr Liz Bagshaw, Editor

---

## Author Response (AR2)

Dear Editor, dear Liz,

many thanks for your detailed instructions and the chance to submit a new revised version! I have worked through all your suggestions and also went through the entire results, discussion and conclusion sections. All suggested updates are made, including figure, as well as all requests for new text, more storytelling and more. There is just one thing we would want to leave as it was: ' L71-78: is this paragraph necessary for your story?' We think that the elastic properties are important, both because we have some viscoelastic modelling included and because ice's elastic/solid nature is the origin of fracture formation. Therefore, we left this paragraph unchanged.

Again, many thanks for all the efforts you have made for improving our manuscript! This is highly appreciated by all of us!

Best wishes,

Angelika and co-authors

---

## Author Response (AR3)

Dear Editor, dear Liz,
many thanks for all the helpful input! Before we present below the point 2 point answers, we'd like to comment on the topic of triangular moulins.

We do agree, most moulin features are not round, but the triangular shape we find in this instance is from our point of view distinct from other drainage fractures. Over the past decade we conducted numerous airborne campaigns over 79NG and Zacharias Isbrae in which we accidentally recorded surface fractures with the airborne camera. All those instances are from their shape different from what we have found at this particular lake. We find very frequent linear fractures with pitted cracks and we also found wide oval shaped drainage locations in areas where the glacier surface is already heavily crevassed (Store Glacier and Zacharias Isbrae). Of course the statistics is still pretty low, as all what we surveyed so far is only a small portion of all lakes, but this particular lake is from out perspective very distinct from others. We do not yet have a very clear explanation what exactly causes the triangular shape, although we found comparable examples in 'surface initiated contact fatigue cracks' (Fig. 11 in https://doi.org/10.1016/j.ijfatigue.2016.12.004). We abstained from discussing such a link before we are more confident, that this is the origin for the triangular shape. We also want to spark the interest in the community to look into their own surveys and data if they find more triangular formed fractures and drainage pathways. This is the reason why we originally suggested this new term gully and why we now want to keep the term triangular in the text. However, we have carefully checked where triangular is absolutely needed and reduced the occurrence by 30%.

Best wishes,
Angelika and co-authors

Revision of 'Supraglacial lake drainage through gullies and fractures'
Thank you for the revised text and the responses to reviewers' comments. One reviewer has provided additional comments on the manuscript that I request that you review and incorporate, along with some minor amendments suggested below.
The reviewer raises the new nomenclature of 'triangular moulins' . This is a tricky one since it was a compromise term. I think we could revert to simply 'moulin' and include a sentence describing the unusual shape, noting the Reviewer's comments on this. Interestingly, some data I am working on seems to suggest our impression of a moulin as a largely 'circular' hole rarely applies, so the triangular shape here may not be that unusual. But if you believe that 'triangular' is important to the explanation, I am happy for it to be retained. A compromise may be referring to either 'triangular feature', 'moulin fracture' or 'moulin' – the shape is how you differentiate it on the imagery, and the moulin/fracture describes the functionality. These terms are used in the text already. The discussion is still a little long, especially after the very detailed results, and even with the attempt to signpost in the first sentence it can be tricky to follow. I wonder if you could either condense it, or add two-three subheadings (Surface fractures, englacial fractures, longer-term changes?) in place of the first sentence? I did find the final paragraph very effective.
I look forward to seeing the final revision.
Liz Bagshaw, Editor

Minor amendments (grammar, sentence structure, readability) to be considered in conjunction with the reviewer's suggestions:
(attention, the line numbers used by the editor are the ones in the diff version)

L16: be consistent with capitalization of southern, northwestern, vs. Northeast. I understand why (NEGIS requires capitalization), but in this sentence I think lower case

We have checked this and "Northeast" should be capitalized as it refers to a specific geographic region.

L16: is the most notable? I would say 'a notable recent example is'
. Could be further simplified: 'While the mass loss was most prominent in the southern and north-western parts in the first decade, it has now reached the northeast of Greenland, where floating tongue of Zacharias Isbræ (ZI) disintegrated in 2013 (REFS)'
Done

L302: remove 'itself' – 'drained' is sufficient
Done.

L305: simplify. 'The 2021 survey (UWB, ALS and Canon camera, 29-07) showed that the moulin had the same horizontal geometry as in 2019, although the triangular feature has snow cover. Laser scanner data…'
Done.

L315: 'are piling up' should be 'piled up'
Done.

L327: replace the new text 'triangular moulins appeared' with 'no new fractures were visible, but the two from 2022 can be…' – since you mention the triangular shape in this sentence, no need to have it in the previous sentence.
Done.

L329 – 332: you have an intro sentence at the end of the previous section ('We will discuss the details of the fracture formation and reactivation further below. ') and then begin the next section with another intro sentence ('Next, we present evidence of crack formation in accordance with lake drainage'). Whilst the signposting is very helpful for the reader, I think just one is sufficient.
We removed the one at the end of the previous section.

L343: delete 'its position is shown as coloured dashed lines' – this information should be in the caption
Done.

L350: Long-winded, rather than telling us what you are going to do, just do it. For example 'A comparison of the drainage in 2005 and 2015 (Figure 4b) shows that the triangular feature can be tracked over the years. In 2015, the crack is 2725 m long and the triangular feature has a side length of 250–300 m. Both features are still well visible in ALOS-2 imagery from 2016-12-18 as surface reflectors , 17 months after their formation. The high-resolution WV imagery from 2020 also shows a triangular-shaped feature of similar size (170–230 m side length), although strongly eroded, and it is still identifiable in 2022, seven year after its formation (Fig. 4). By 2021,
the crack-line visible in 2015 has disappeared, and the triangular feature is slightly off the trajectory between regions A–B, 200–500 m further south. Most importantly, it now sits at higher elevation, at the margin of the lake's middle branch, and is thus unlikely the main drainage channel. '
We tried to followed the previous advice not to jump straigth into, but to introduce the next step with an intro. We are happy to follow the new advice and have changed this paragraph accordingly.

L364-365: a stranded sentence, suggest joining to the previous paragraph.
Done.

L366: 'The crosses in Fig. 8 denote principal stress directions'
– this information is in the caption so can be removed from the main text. Instead 'The cracks predominantly match the principal stress directions (calculated by inverse modelling, as in REF, Fig. 8), indicating the dominant fracture mode is tensile (mode I). Close inspection of the surface of the lake ground shows narrow and potentially shallow cracks to exist in the area inside and outside the lake, and also
the presence of cracks that are not aligned with the principle direction (Fig. 8d).
We have changed this paragraph accordingly.

L370-375 is a little confusing. Suggest: 'This is explained by the formation of tiny cracks at a 45° angle, which are formed in the main shear direction (similar to Humbert et al. 2023). Once these cracks propagate at 45°, they form a connection from one row of narrow cracks to the next, where the crack propagation follows the main principle direction for a while until it again jumps onto the 45° and so on'
Many thanks for the suggestion! We have changed this paragraph accordingly.

L403: Agree with reviewer that 'Englacial features' or 'Englacial signals' would be a better title for this section
We have changed it to 'Englacial features'

L410: lake base rather than 'lake ground'?
Yes, lake base is also fine. In the literature also lake bottom is used.

L426-428: could be removed without detriment, or refer to 'T1' in the text to relate directly to figure
We have removed this text.

L441: tenses not quite correct in this new sentence: 'These radargrams give some insight into englacial drainage pathways, so now we must consider where water masses might be stored beneath the glacier, and how this varies in time' . It should also be joined to the next paragraph, rather than left stranded here.
We have corrected the tense in the sentence and moved it to the next paragraph.

L480: 'The discussion is structured by' rather than 'we are structuring the discussion' . Suggest 'We begin our discussion with an evaluation of the evolution of the lake, followed by an analysis of the surface and englacial fractures, before attempting to understand their long-term change and the climate context. ' , or including subheadings as suggested above.
We have considered the subheadings, but the subsections would be so imbalanced in length, so we decided against this.

L483: 'areal extent remained lower' – unclear whether this means that the lake area is lower in elevation, or whether the surface area extent is reduced.
Has been changed to 'the size of the lake remained lower'

L484: unclear what is meant by 'lower load'
We have rephrased this to 'lower mechanical load'

L485: sentence is very confusing: 'the fractures are for longer (does this mean longer in time or space?) in the topographic low and with that in the area of highest water pressure at the lake ground' – can you rephrase this?
Indeed 'for a long time' is the correct statement. We have changed this accordingly.

Reviewer comments

General Comments

17 years after Das, the title 'Supraglacial lake drainage through fractures' is perhaps overly general and not good at communicating what is unique about this study and its contribution to the literature. Is there a better title that can communicate the multi-modal and multi-annual/long-term nature of the observations?
This is a good point. We have found a new title.

In the response to reviewers and in the text, there is now a focus on 'triangular moulins'. I am slightly confused as to why the triangular component is significant here. It seems to imply that there is something distinct and important about the triangular shape that is not found in alternatively-shaped drainage moulins (Das, Doyle, Chudley, etc). These other moulins are not triangular but are still distinctly angular when fresh. Is it not just a coincidence as to how the final shape appears? It is interesting that multiple generations of moulins from this lake appear to be triangular, but this manuscript doesn't appear to address (i) what might enforce this; and (ii) why it is important or distinct cf. other shapes. My guesses are that either the triangle forms due to the intersection of a second fracture, orthogonal to the first (Fig 4a, 8a, 9) or that ongoing water input into the moulin thermomechanically erodes a gorge-like feature orthogonal to the initial crack (Fig 5a, 7a, 8c/d, etc). But I'm not sure whether this is particularly important to necessitate any distinct terminology.
There has always been a focus on the triangular shape of the moulins, but they were denoted by us gullies. This was not well received by the reviewers and we had to change this to standard terminology and this is why triangular moulins, triangular moulin fractures appear in this version. Indeed, the shape is distinct – very distinct. Which is why we wanted to introduce a new term. No, it is not just a coincidence. We have surveyed numerous other lakes and in none of those cases a similar shape is found. Just last summer, we surveyed the lake in Chudley's paper and the fracture pattern is very different.

Minor Comments (attention, the line numbers used by the reviewer are the ones in the new manuscript not the diff version)

L17 - "Zacharias Isbræ" - why not Zachariae Isstrøm, as is more common in the literature? Interestingly, some light googling appears to suggest that most references to Zacharias Isbræ come from literature originating from this group. I am genuinely open and interested to know if there is something this group knows with regards to the proper naming of ZI that others do not - certainly, this has happened in the past!
We have no background in linguistics and happy to change this to Zachariæ Isstrøm (done). It definitely obeys more characteristics of an outlet glacier (bræ) then an ice stream (isstrøm), but if the community likes to call an outlet glacier an ice stream, we won't object this. In this case, it would be worth to consider the actual correct form for Zachariae written as Zachariæ.

L40 - 'tenths of metres' - perhaps 'tens of metres'?
Done.

L76 - 'mumulti-chromatic0.3'
Done.

L78 - 'screened' - perhaps 'manually quality-assessed, with poor-quality/cloudy images

removed' for clarity?
We have rephrased screened.

L79 - Was the SNAP toolbox actually used by the authors, or was for processing an existing data source (Copernicus Browser, Google Earth Engine, etc.)? If the latter, perhaps it's preferable to simply state where the data was sourced.
Yes, the SNAP toolbox was used by the lead author.

L132 - Under section 2.3 (and before section 2.3.1), it might be nice to have a top-level summary stating general information about the aircraft mission, flight paths, flight dates, etc.
We inserted some top-level summary, although this is counter productive in terms of shortening the manuscript and the request to make this easier to read. The dates of the relevant survey flight were anyway given in the respective figures. We don't agree to insert another figure as in the overview figure 1 the profile sections of the relevant radar data is shown. We refer to the online AWI radar data viewer, were all profile lines are shown.

L134, 148, 149, and perhaps elsewhere - repeated instances of 'gegeoref' as an error. A find-and-replace issue, perhaps?
Apologies - corrected.

L216 - 'Ice-Sheet' > 'Ice-sheet', and perhaps also include 'ISSM' in brackets, as an acronym many might be more casually familiar with.
Done.

L235-236 - 'One can constrain the initiation of this lake' - perhaps reword as 'No lake was observed to form in the observational data until…', which more explicitly states the evidence (if this is indeed the case)
We have rephrased this to 'Although the data coverage in the 1980's and 1990's is sparse, no lake was observed to form in the observational data until 1995.'

L239-240. Was there no observed filling between 2007-2011? This might also be worth stating explicitly..
We even have a table (Tab. 2) that states this explicitly.

Fig 4 - I had assumed each of the sequential colours in panel b is indicative of one year passing, but this isn't the case (there are only enough lines in the viridis colorscheme to get us to 2020). Perhaps these need individually labelling?
The lines are now labelled with a date.

Fig 8 - RE: my previous comment about coloring the crosses. As someone color-deficient myself, I appreciate that the author's concern for color-blind interpretability, but my critique was primarily about struggling to differentiate the contrast of the two directions. If the authors do not wish to switch to colors, would is be possible to make the darker cross slightly darker and the lighter cross slightly lighter?
We have changed the crosses into a coloured version.

L357 - and elsewhere. This is a long paper - currently on page 23! - that incorporates many methods, and by this point I was beginning to lose track of the purpose of some of the approaches. The purpose of the modelling is not introduced in the intro, methods, nor here, so I was at a loss as to why it would be important to 'evaluate the shape and size stability of englacial channels'. Although it might be strictly considered discussion, I wonder whether it would be possible to have a small introductory remark (e.g. "In response to [...] it is necessary to establish [...]. Hence, we…"). I would recommend this for all sections, not just this one - the next section (3.4 Moulin refilling) was another moment where I found myself

wondering why this analysis was being done. Although the discussion should make all this clear, in this longer paper some additional 'signposting' would be beneficial to keep the reader on track! (L430/Section 3.7 actually does this well).
We do not agree here. The purpose of modelling is clearly stated in the introduction: 'We incorporate viscoelastic modelling as a case study to understand if and how drainage pathways close over time.' and moreover the type of modelling is introduced in an entire paragraph, that was even considered to be not necessary by reviewers before.

Section 3.5 - This section is called 'Englacial channels', but nowhere in the text are the englacial features identified as channels. Is there anything that could be made explicit that it is correct to interpret these as channels and not as e.g. (water-filled) fractures, or static perched englacial water bodies?
We have renamed the section to Englacial features.